# The molecular basis of socially induced egg-size plasticity in honey bees

Bin Han[1,2†], Qiaohong Wei[1†], Esmaeil Amiri[2,3†], Han Hu[1], Lifeng Meng[1], Micheline K Strand[4], David R Tarpy[5], Shufa Xu[1], Jianke Li[1]*, Olav Rueppell[6]*

[1]Institute of Apicultural Research/Key Laboratory of Pollinating Insect Biology, Ministry of Agriculture and Rural Affairs, Chinese Academy of Agricultural Sciences, Beijing, China; [2]Department of Biology, University of North Carolina Greensboro, Greensboro, United States; [3]Delta Research and Extension Center, Mississippi State University, Stoneville, United States; [4]Biological and Biotechnology Sciences Branch, U.S. Army Research Office, DEVCOM-ARL, Baltimore, United States; [5]Department of Applied Ecology, North Carolina State University, Raleigh, Canada; [6]Department of Biological Sciences, University of Alberta, Edmonton, Canada

*For correspondence:
apislijk@126.com (JL);
olav@ualberta.ca (OR)

†These authors contributed equally to this work

Competing interest: The authors declare that no competing interests exist.

**Abstract** Reproduction involves the investment of resources into offspring. Although variation in reproductive effort often affects the number of offspring, adjustments of propagule size are also found in numerous species, including the Western honey bee, *Apis mellifera*. However, the proximate causes of these adjustments are insufficiently understood, especially in oviparous species with complex social organization in which adaptive evolution is shaped by kin selection. Here, we show in a series of experiments that queens predictably and reversibly increase egg size in small colonies and decrease egg size in large colonies, while their ovary size changes in the opposite direction. Additional results suggest that these effects cannot be solely explained by egg-laying rate and are due to the queens' perception of colony size. Egg-size plasticity is associated with quantitative changes of 290 ovarian proteins, most of which relate to energy metabolism, protein transport, and cytoskeleton. Based on functional and network analyses, we further study the small GTPase Rho1 as a candidate regulator of egg size. Spatio-temporal expression analysis via RNAscope and qPCR supports an important role of *Rho1* in egg-size determination, and subsequent RNAi-mediated gene knockdown confirmed that *Rho1* has a major effect on egg size in honey bees. These results elucidate how the social environment of the honey bee colony may be translated into a specific cellular process to adjust maternal investment into eggs. It remains to be studied how widespread this mechanism is and whether it has consequences for population dynamics and epigenetic influences on offspring phenotype in honey bees and other species.

## Editor's evaluation

This study provides valuable insights into the control of egg size plasticity, a key form of maternal investment. It presents convincing evidence from both experimental manipulations and molecular investigations of egg plasticity in honey bee queens. It will be of interest to evolutionary biologists, particularly those working on life-history trade-offs and reproductive strategies.

## Introduction

Life history trade-offs are a pervasive attribute of life, and how organisms balance growth, reproduction, and survival governs much of their evolution (*Stearns, 1989*; *Flatt, 2020*; *Bonsall et al., 2004*). Simultaneous optimization of all traits is impossible as all individuals are limited by time,

**eLife digest** Honey bees are social insects that live in large colonies containing tens of thousands of individuals. The vast majority of bees are sterile females known as worker bees. They perform most of the activities essential for the survival of the colony, including foraging for pollen and nectar and taking care of eggs and larvae.

An individual known as the queen bee is the mother of the colony and is normally the only female who reproduces. She has two massive ovaries and can produce up to two thousand eggs per day. Previous studies indicate that the number and size of the eggs vary according to the conditions inside the colony and in the surrounding environment. Larger eggs contain more nutrients so the resulting embryos may have a better chance of survival. However, producing bigger eggs requires the queen to invest more resources, which is costly to the colony as a whole.

It remains unclear which mechanisms regulate the size of honey bee eggs. To address this question, Han, Wei, Amiri et al. carried out a series of experiments on the Western honey bee, *Apis mellifera.* The experiments showed that queen bees in small colonies had smaller ovaries and produced bigger eggs than those in large colonies. The difference in egg size appeared to be due to the queen bee's perception of the size of the colony, rather than its actual size.

An approach called proteomics revealed that 290 ovarian proteins were produced at different levels in big-egg producing ovaries compared to small-egg producing ovaries. Further experiments suggested that a protein known as Rho1 regulates the size of the eggs the queen bees produce.

These findings provide an explanation for how the social environment of the Western honey bee colony may influence the queen bee's reproductive investment at the molecular level. Further studies to confirm and expand on this work may help to improve honey bee health and also contribute to our general understanding of this life stage in bees and other insects.

resources, or other factors (*van Noordwijk and de Jong, 1986*). Optimization of offspring provisioning has resulted in a wide variety of reproductive strategies that are characterized by species-specific trade-offs between offspring size and number. Numerous studies have analyzed a possible trade-off between offspring number and size across and within many different species (*Smith and Fretwell, 1974*; *Dani and Kodandaramaiah, 2017*; *Fox and Czesak, 2000*; *Berrigan, 1991*; *Church et al., 2021*), while social insects have received little attention despite the exceptional reproductive specialization of their female castes (*Church et al., 2021*). The topic has gained additional interest due to its implications for inter-generational effects that can profoundly affect organismal phenotypes and evolutionary dynamics (*Rasanen and Kruuk, 2007*; *Sánchez-Tójar et al., 2020*; *Bebbington and Groothuis, 2021*). Environmental conditions typically lead to plastic responses, most often in the form of variation in offspring number. However, propagule size evolves according to life history and environmental selection (*Church et al., 2019*) and can also be flexibly adjusted, particularly in plants and insects (*Dani and Kodandaramaiah, 2017*) and in some bird species (*Christians, 2002*). While the fitness consequences of egg-size variation have been studied extensively (*Fox and Czesak, 2000*; *Azevedo et al., 1997*), little is known about the proximate regulation of egg size, which is equally important for our understanding of this fundamental, complex life history trait (*Flatt, 2020*; *Jha et al., 2015*).

Social evolution changes selection pressures and adaptive evolution due to kin selection (*Abbot et al., 2011*), particularly in eusocial insects with colonies that form a distinct level of selection (*Boomsma and Gawne, 2018*). In these species, many individuals contribute to a homeostatically regulated colony environment (*Oster and Wilson, 1978*), pronounced phenotypic plasticity results in individuals specialized for particular functions (*Flatt et al., 2013*), and resource transfers among kin influence reproductive value (*Lee, 2003*). Thus, life history evolution in eusocial insects differs fundamentally from that of other species (*Shik et al., 2012*; *Negroni et al., 2016*) and has generated some extraordinary trait combinations that defy traditional life history trade-offs (*Keller and Genoud, 1997*). Specifically, the reproductively specialized queen caste is typically well-provisioned and cared for by nonreproductive workers, which also perform all of the intensive brood care that is characteristic of eusocial insects. As such, social insect queens may not be resource-limited despite their very high reproductive effort (*Schrempf et al., 2017*). Nevertheless, honey bee (*Apis mellifera*) queens display

plasticity in egg size that is consistent with patterns in solitary species and corresponds to an adaptive investment hypothesis *Hall et al., 2020*; egg size is relatively small under favorable conditions, such as food abundance and a large colony (>6000 workers), and is relatively large when food availability or colony size declines (*Amiri et al., 2020*). As in other species (*Fox and Czesak, 2000*), larger eggs have been associated with a survival advantage for the resulting honey bee worker offspring (*Amiri et al., 2020*). One honey bee queen typically serves as the sole reproductive in her colony regardless of colony size. The queen is fed and cared for by her workers, although it is unknown how much food she receives and how queen care changes with colony size. Only one, native queen is typically found in a honey bee colony, but queens can be experimentally transferred between colonies, although some may be rejected and killed during this process. Queen condition also affects egg size (*Al-Lawati and Bienefeld, 2009*), and egg size differs between worker- and queen-destined eggs (*Wei et al., 2019*) with important consequences for caste determination (*Al-Kahtani and Bienefeld, 2021*) and queen reproductive potential (*Yu et al., 2022*). However, none of these phenomena has been explored further to understand the underlying mechanisms that govern variation in egg size.

Here, we report our findings of an in-depth investigation of how egg-size plasticity in honey bee queens is regulated and contribute knowledge of the molecular regulation of insect egg size in general, which has been difficult to determine because hundreds of genes may be involved (*Jha et al., 2015*). Our results demonstrate that the previously identified effects of colony size on queen egg size (*Amiri et al., 2020*) are reversible and not fixed. We further establish that egg size is actively regulated and not simply a passive consequence of egg-laying rate, although egg size and egg-laying rate can be negatively correlated. This correlation may explain why smaller ovaries can produce larger eggs. We further show that the social cue triggering changes in egg size within the queen does not require physical contact. Proteome comparisons between queen ovaries producing small versus large eggs indicate a central role of protein localization and cytoskeleton organization. We additionally demonstrate that the knockdown of the central cytoskeletal regulator *Rho1* significantly decreases egg size. Our data thus suggest that social cues can be translated into specific molecular processes to control plastic reproductive provisioning, which presumably evolved as an adaptation to the colonial life cycle of honey bees.

## Results

### Honey bee queens reversibly adjust egg size in response to colony-size changes

The first experiment involved repeated transfers of queens among colonies of different sizes, which was designed to expand our previous findings that honey bee queens can regulate their egg size in response to colony conditions (*Amiri et al., 2020*). Sister queens that were housed in medium-sized colonies at the start of our first experiment produced a range of intermediate egg sizes with significant inter-individual differences (ANOVA: $F_{(10,219)} = 31.5$, p<0.001). Over the course of the first week, egg sizes significantly increased (t = 5.8, df = 10, p<0.001) while egg number did not differ significantly (t = 0.7, df = 10, p<0.482). The first and second measurements were correlated for egg size ($R_P = 0.80$, n = 11, p=0.003), indicating consistent differences among queens, but not egg number ($R_P = 0.31$, n = 11, p=0.350). After transfer from medium to small colonies, the egg size of all six queens increased significantly (for each queen: $F_{(1,38)} = 23.7–153.3$, p<0.001). In contrast, egg size significantly decreased for all five queens that were transferred from medium to large colonies ($F_{(1,38)} = 8.9–53.2$, all p<0.005). Our reciprocal transfers after the fourth week showed that egg-size adjustments were reversible because all three queens successfully transferred from large to small colonies significantly increased their egg sizes ($F_{(1,38)} = 143.8–1001.8$, all p<0.001) and all five queens transferred from small to large colonies significantly decreased the size of their eggs ($F_{(1,38)} = 123.0–699.4$, all p<0.001). The egg size of most queens did not significantly change between separate measures in the same-sized colonies (third versus fourth or fifth versus sixth week). While egg size and number were not significantly correlated while all queens were housed in medium colonies, there was a negative correlation between size and number in weeks 3–6, although this relation was only significant in week 3 ($R_P = –0.83$, n = 11, p=0.002), week 5 ($R_P = –0.95$, n = 8, p<0.001), and week 6 ($R_P = –0.90$, n = 8, p=0.003). Thus, honey bee queens consistently adjust the number and size of their eggs in response to colony size despite inter-individual differences in egg size (*Figure 1*, *Figure 1—source data 1*).

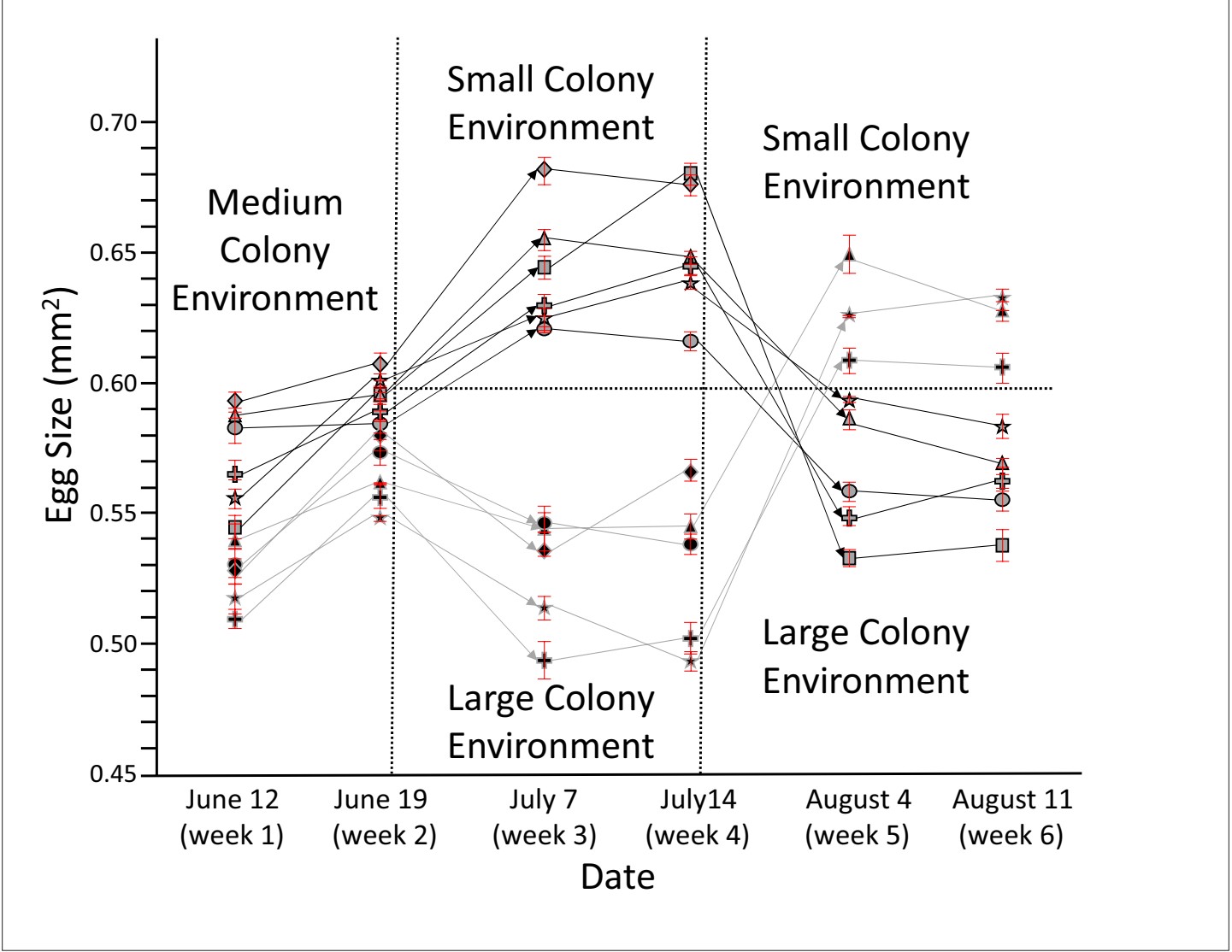

**Figure 1.** Honey bee queens reversibly adjust egg size according to colony size. The egg size (n = 20 for each data point) of individual queens (each depicted by a unique symbol ± SEM) was measured for 6 weeks while they were moved from medium to small to large (light symbols with dark lines) or from medium to large to small colonies (dark symbols with light lines). Arrows symbolize transfer between different colonies, during which some of the queens died (lines not continuing). Despite the presence of individual and environmental differences, these experiments show a strong and consistent negative relation between egg size and colony size. Ovaries from surviving queens were collected after week 6 for proteomic profiling (see below).

The online version of this article includes the following source data for figure 1:

**Source data 1.** Egg-size measurements.

The surviving queens of this experiment, plus two additional large-egg-producing queens to increase sample size (*Figure 1—source data 1*), were compared with regard to size, body weight, and ovary weight. Queens in small colonies had significantly lighter ovaries than those in large colonies ($F_{(1,8)}$ = 10.2, p=0.013), while body size ($F_{(1,8)}$ = 0.3, p=0.596) and wet weight ($F_{(1,8)}$ = 0.8, p=0.402) did not differ (*Figure 2A*, *Figure 2—source data 1*). These results were confirmed in a second comparison between queens that were simply housed in small versus large colonies after maturation (ovary: $F_{(1,6)}$ = 28.7, p=0.01; size: $F_{(1,6)}$ = 0.07, p=0.805; weight: $F_{(1,6)}$ = 0.3, p=0.627; *Figure 2B*, *Figure 2—source data 2*). A third comparison between queens housed in small versus large colonies indicated that similar-sized queens ($F_{(1,10)}$ = 0.1, p=0.748) can differ not only in ovary weight ($F_{(1,10)}$ = 18.5, p=0.002) but also in body weight ($F_{(1,10)}$ = 5.6, p=0.039; *Figure 2C*, *Figure 2—source data 3*). All egg-size measurements are shown in *Supplementary file 1*. Across these three experiments, no significant

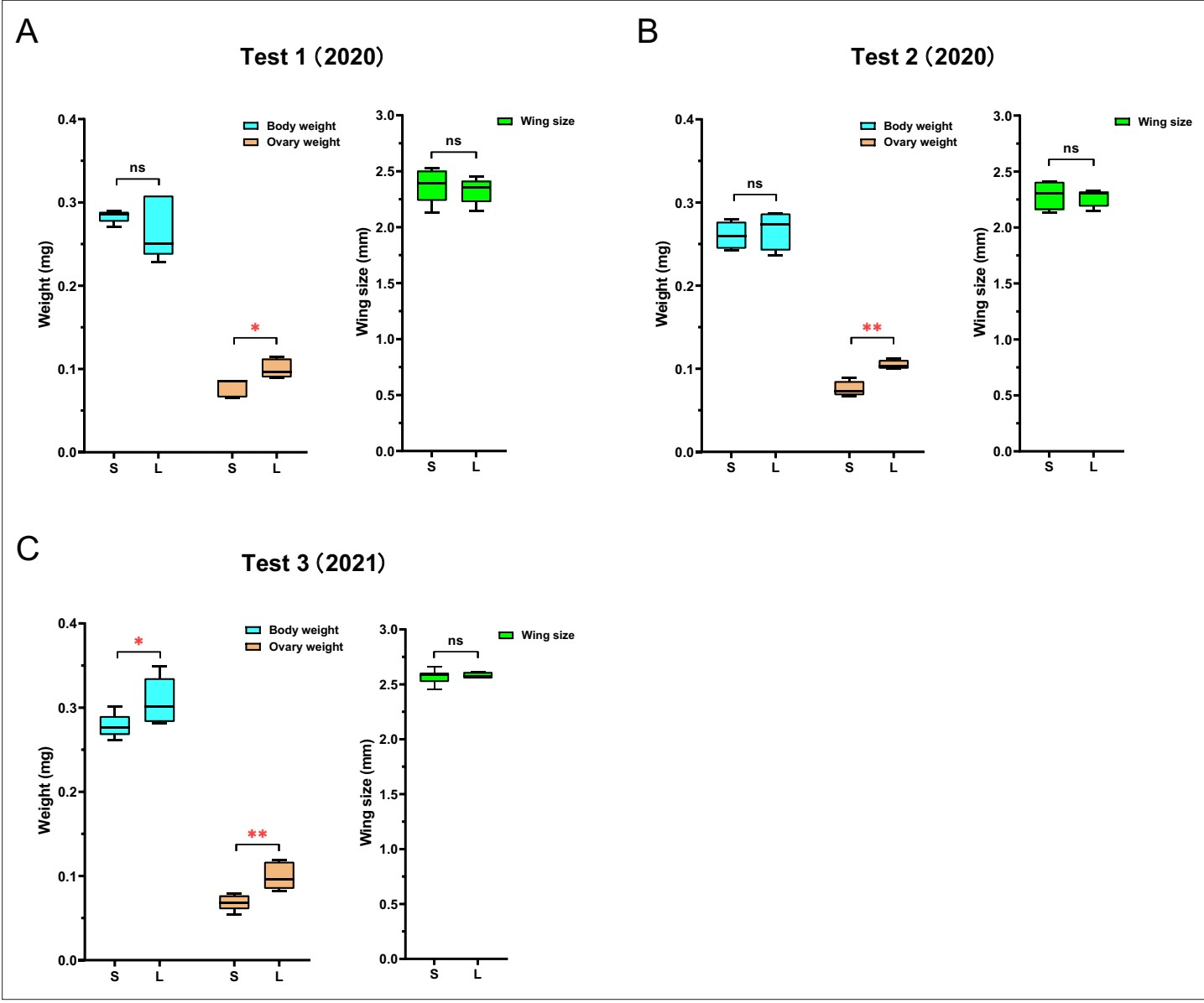

**Figure 2.** The queens' ovary weighs less in 'small' colonies than in 'large' colonies. While queen size, measured as wing size, was not significantly different between queens in 'large' (L) and 'small' (S) colonies, ovaries were consistently lighter in queens from small colonies than in queens from large colonies. Total body weight of queens showed no significant difference between the two groups in the first (**A**) and second (**B**) experiments, but queens in large colonies were significantly heavier than queens in small colonies in the third experiment (**C**). $N_{Test1}$ = 10, $N_{Test2}$ = 8, $N_{Test3}$ = 12 for all measures; simple ANOVAs were used for pairwise comparisons (*p<0.05, **p<0.01).

The online version of this article includes the following source data for figure 2:

**Source data 1.** Queen measurements for test 1.

**Source data 2.** Queen measurements for test 2.

**Source data 3.** Queen measurements for test 3.

relationships between queen size and egg size were found for queens in small ($R_P$ = 0.11, n = 15, p=0.702), medium ($R_P$ = 0.21, n = 8, p=0.616), or large ($R_P$ = –0.22, n = 15, p=0.440) colonies.

## Egg size is unaffected by an experimental hiatus in egg laying

In our first experiment, we found that egg size was negatively correlated to egg number produced. To test whether small egg size is merely a passive consequence of high egg-laying rate, we thus assessed egg size before and after a 2-week period of queen caging, which prevented queens from laying any

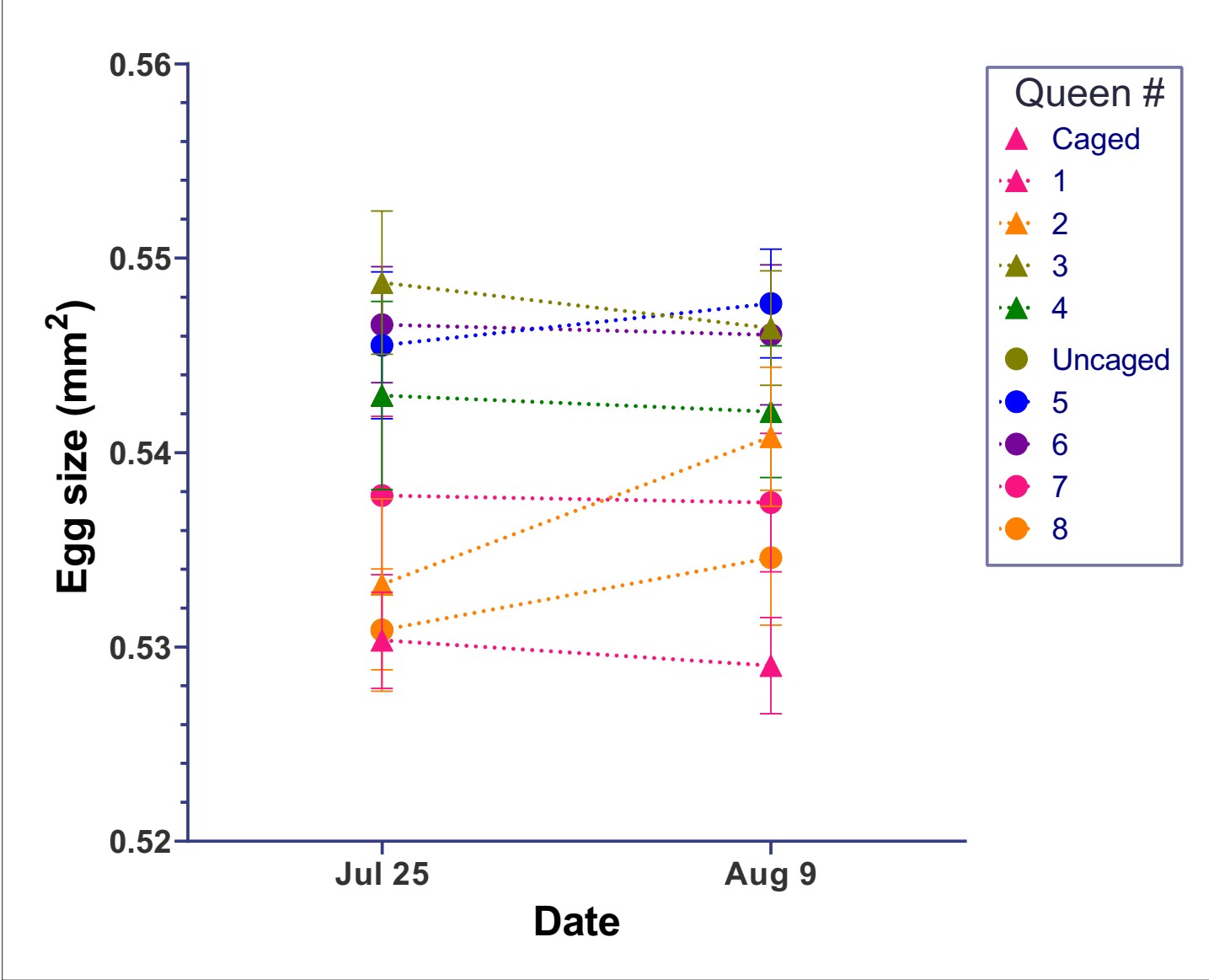

**Figure 3.** Egg size of queens is not affected by egg-laying rate. After egg size of individual queens in large colonies was measured, treatment queens (triangular symbols) were confined on capped brood comb that did not allow any oviposition while the control queens (circle symbols) had free access to empty comb for oviposition. After 14 days, the egg size in neither group of queens changed significantly. Individual means ± SEM of 20 eggs are shown for each summary data point.

The online version of this article includes the following source data for figure 3:

**Source data 1.** Egg-size measurements.

eggs. None of the four caged queens significantly changed her egg size ($F_{(1,38)}$ = 0.02–1.8, all p>0.1). None of the four queens in an unmanipulated control group during the same time changed her egg size either ($F_{(1,38)}$ = 0.005–0.6, all p>0.4), and egg sizes were similar between the restricted and unrestricted groups overall (*Figure 3*, *Figure 3—source data 1*).

## Queens adjust their egg size in response to perceived instead of actual colony size

To better understand how colony size influences queen egg-size regulation, the perceived but not the physical colony size of small colonies was extended. The queens in 'small' colonies, producing relatively large eggs, were paired via a double-screened tunnel with medium-sized hive boxes that either

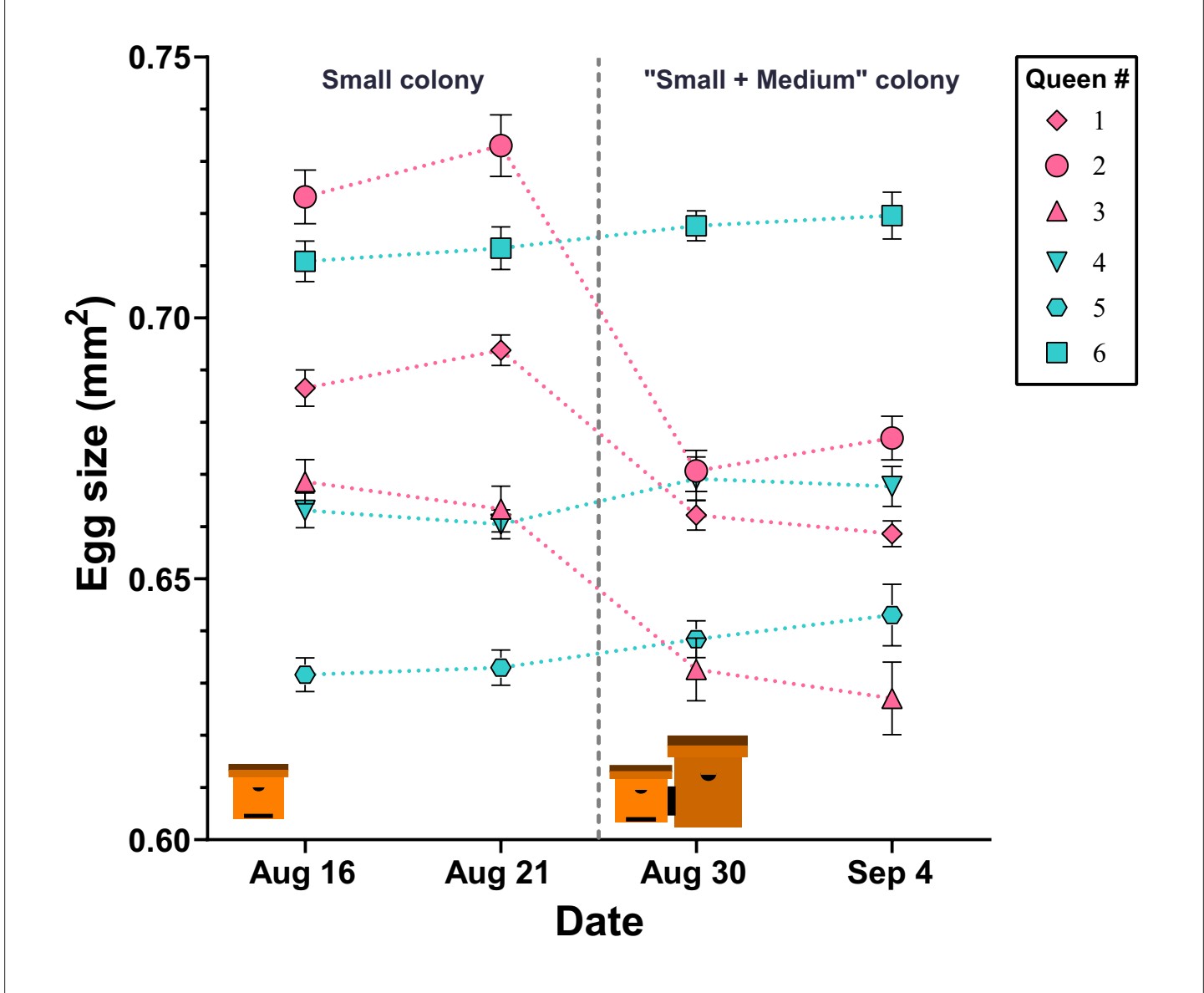

**Figure 4.** Egg size is actively regulated by the queen in response to perceived colony size. After initial egg-size determination, queens in 'small' hives were either paired with an empty 'mMedium' hive box (controls: cyan color, #4, #5, and #6) or with a 'medium' hive box containing a colony (pink color, #1, #2, and #3). Queens in hives that were paired with another colony decreased their egg size, while queens in control colonies maintained their egg sizes. Individual means ± SEM are shown.

The online version of this article includes the following source data for figure 4:

**Source data 1.** Egg-size measurements.

contained empty frames or a queenless, 'medium' colony. All three queens paired with a regular colony reduced the size of their eggs compared to their initial egg size (Q1: $F_{(3,76)}$ = 34.5, p<0.001; Q2: $F_{(3,76)}$ = 42.5, p<0.001; Q3: $F_{(3,76)}$ = 14.6, p<0.001; post-hoc tests indicated significant differences only between measurements before and after manipulation; *Figure 4*, *Figure 4—source data 1*). In contrast, none of the three control queens significantly changed their egg size during the experimental period (Q1: $F_{(3,76)}$ = 1.3, p=0.297; Q2: $F_{(3,76)}$ = 1.6, p=0.196; Q3: $F_{(3,76)}$ = 1.0, p=0.379; *Figure 4*, *Figure 4—source data 1*).

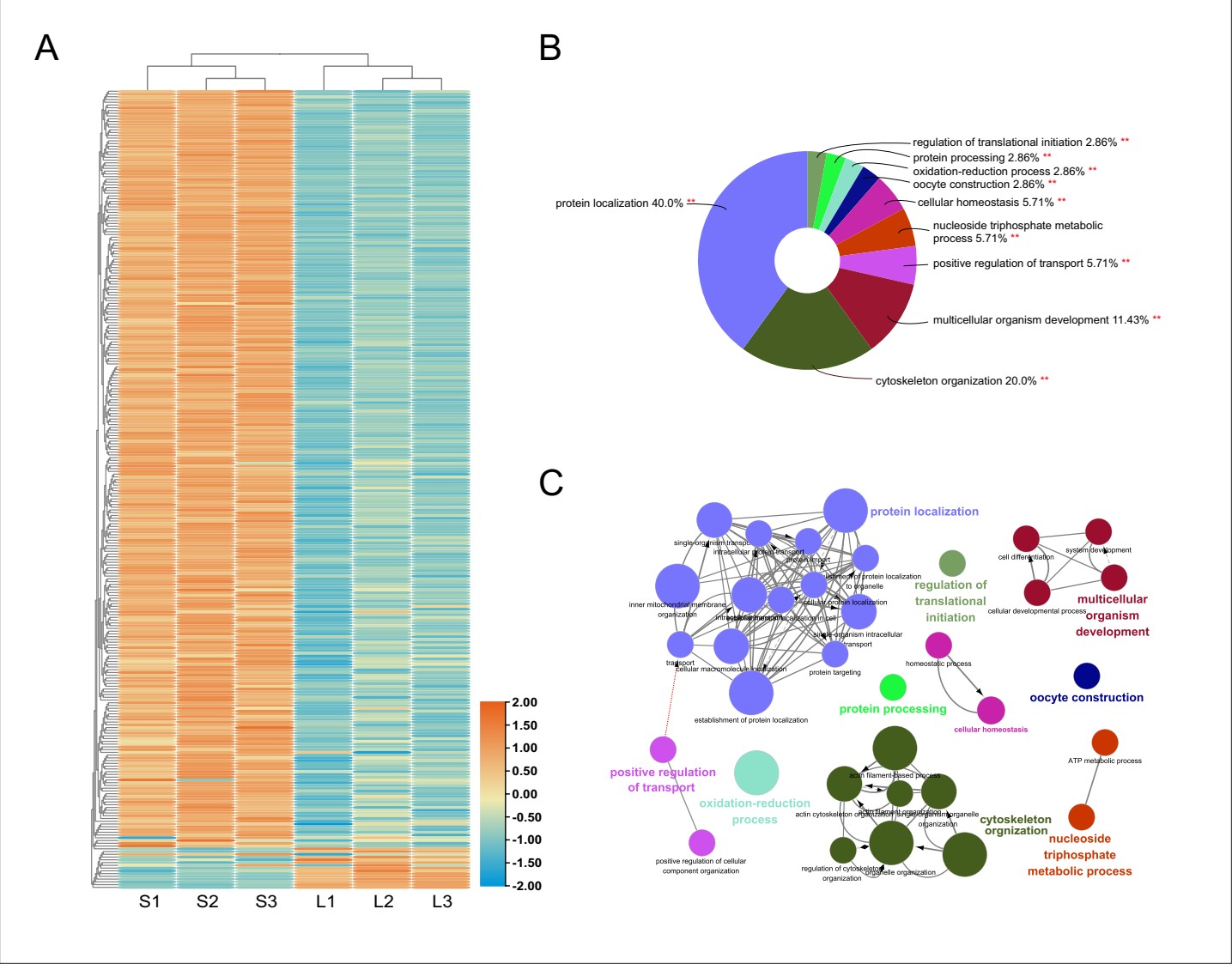

**Figure 5.** Quantitative protein differences between the ovaries of queens producing small and large eggs. The abundance of approximately 10% of all identified proteins was significantly different, with the vast majority of significant differences indicating higher protein levels in the ovaries of queens that produced larger eggs because they were housed in small instead of large colonies (**A**). Among the Gene Ontology (GO) terms that were significantly (p<0.01) enriched in the differentially abundant proteins, 'protein localization' and 'cytoskeleton organization' were most prominent (**B**). Functional grouping of these overall GO terms, using kappa ≥ 0.4 as linking criterion confirmed that the GO terms represented at least six distinct functional groups (**C**).

## Ovary proteome comparisons suggest that egg size is increased by cellular transport and metabolism

To compare the ovary proteome of queens producing large eggs with that of queens producing small eggs, we identified a total of 2022 proteins and compared their relative abundance. Among the 290 proteins that exhibited significant quantitative differences, significantly more proteins were more abundant (275) than less abundant (15) in large-egg-producing ovaries compared to small-egg-producing ovaries ($\chi^2$ = 233.1, p<0.001; *Figure 5A*, *Supplementary file 2*).

Gene Ontology (GO) analysis of the proteomic data showed that the proteins with increased abundance in large-egg-producing ovaries compared to small-egg-producing ovaries were significantly enriched in 10 biological process terms (*Figure 5B and C*, *Supplementary file 3*): 'protein localization' (p=0.00004), 'oxidation-reduction process' (p=0.00006), 'cytoskeleton organization' (p=0.00009), 'cellular homeostasis' (p=0.0003), 'protein processing' (p=0.006), 'nucleoside triphosphate metabolic

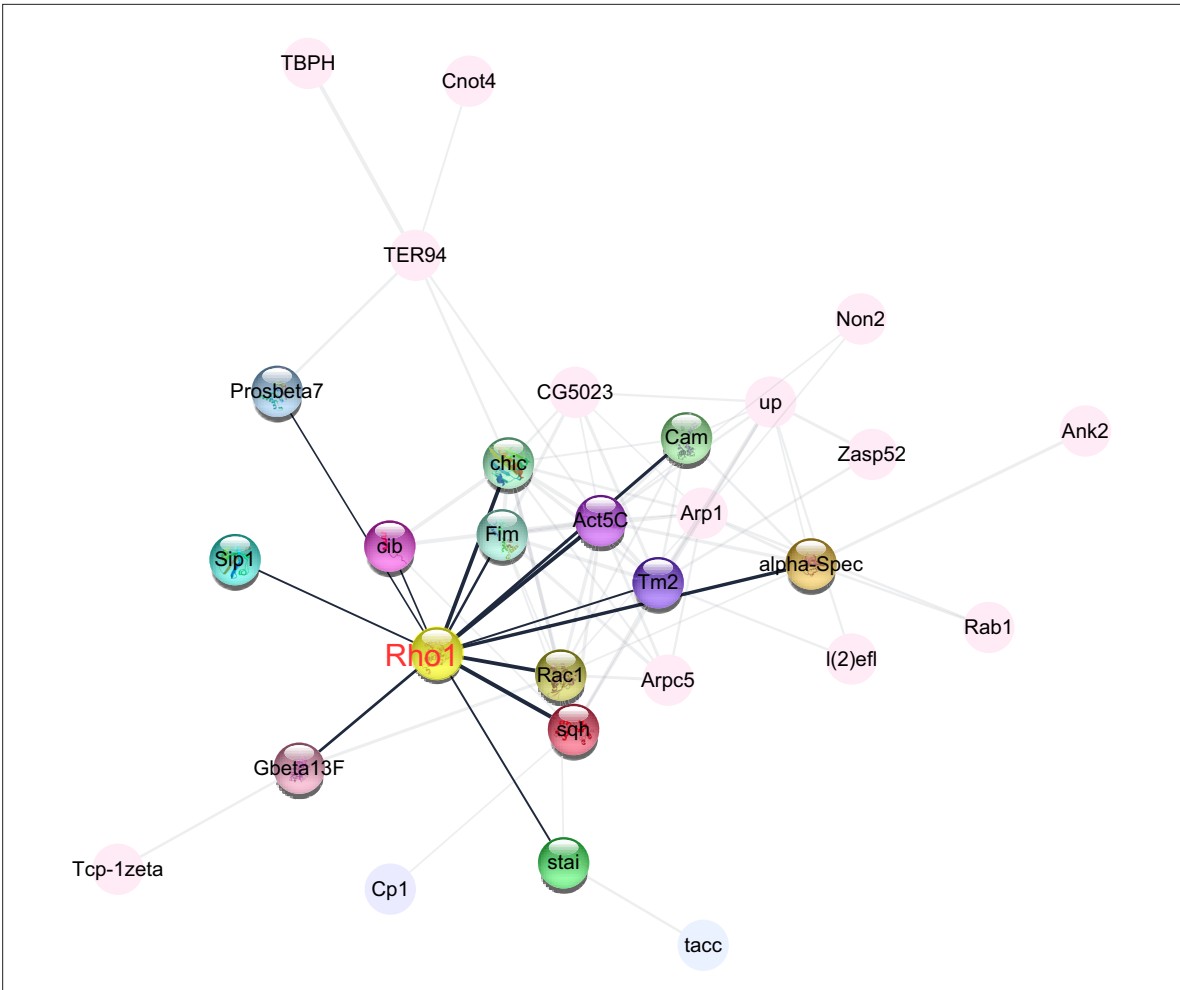

**Figure 6.** Central role of Rho1 in protein–protein interaction network of cytoskeleton organization, a Gene Ontology (GO) term that was enriched among proteins that were more abundant in the ovaries of large-egg-producing honey bee queens than in small-egg-producing queens. The interaction analysis, carried out in STRING v10, linked 29 proteins into the network. The highlighted nodes depict proteins that have a direct interaction with Rho1, a central regulator of cytoskeletal organization and the second most connected protein in the network.

process' (p=0.009), 'positive regulation of transport' (p=0.010), 'multicellular organism development' (p=0.011), 'oocyte construction' (p=0.046), and 'regulation of translational initiation' (p=0.048). In contrast, no GO term was significantly enriched in the proteins that were less abundant than in small-egg-producing ovaries.

The KEGG analysis revealed an enrichment of seven key pathways in the proteins with increased abundance in large-egg-producing ovaries (*Supplementary file 4*), which included 'glycolysis/ gluconeogenesis' (p=0.00002), 'citrate cycle (TCA cycle)' (p=0.00003), 'RNA transport' (p=0.0003), 'beta-alanine metabolism' (p=0.0005), 'protein processing in endoplasmic reticulum' (p=0.0005), 'proteasome' (p=0.0008), and 'oxidative phosphorylation' (p=0.004). Consistent with the GO analysis, no enrichment could be identified in the proteins that were less common in large-egg-producing ovaries compared to small-egg-producing ovaries.

The two largest GO term categories were 'protein localization' and 'cytoskeleton organization.' Of the 34 differentially expressed proteins that were associated with 'cytoskeleton organization,' 29 were connected by a protein–protein interaction (PPI) analysis (*Figure 6*, *Supplementary file 5*). This analysis pointed to five proteins with >10 connections to other proteins: Act5C (15), Rho1 (13), chic (12), Rac1 (11), and Tm2 (11). Instead of Act5C, the most connected protein with essential structural functions (*Wagner et al., 2002*), we decided to further investigate the role of the second most connected protein Rho1, which represents a key regulator of cytoskeletal organization (*Kim et al., 2018*).

## Rho1 in ovaries plays an important role in egg-size regulation

Based on our proteomics results and functional evaluation of the top candidate genes, we hypothesized that *Rho1* is important for egg-size regulation. RNAscope in situ hybridization enabled a fine-scale characterization of *Rho1* expression in the ovary, which was consistent with this hypothesis; little *Rho1* was expressed in the terminal filament but some expression was discernible in the germarium, concentrated in the cytocyst (incipient oocyte). Relative strong expression of *Rho1* was found in the growing oocytes of the vitellarium in contrast to nurse and follicle cells at that developmental stage. In mature oocytes, *Rho1* expression was again low (**Figure 7A**). In the oocytes, *Rho1* was mainly located near the lateral cell cortex, which may represent areas of longitudinal growth (**Figure 7B**).

RNAi-mediated knockdown of *Rho1* resulted in an average of 35.1% reduced *Rho1* expression compared to controls (**Figure 8A**, **Figure 8—source data 1**). Expression of *Rho1* was also on average 57.0% higher in control queens from small colonies that produce large eggs than queens from large colonies that produce small eggs (**Figure 8A**). The knockdown of *Rho1* consistently decreased egg sizes (**Figure 8B**, **Figure 8—source data 2**) in all three queens in small colonies (Q10: $F_{(1,38)} = 177.8$, $p<0.001$; Q11: $F_{(1,38)} = 139.7$, $p<0.001$; Q12: $F_{(1,38)} = 44.6$, $p<0.001$) and large colonies (Q4: $F_{(1,38)} = 63.7$, $p<0.001$; Q5: $F_{(1,38)} = 42.8$, $p<0.001$; Q6: $F_{(1,38)} = 28.1$, $p<0.001$), while none of the six corresponding control queens exhibited significant egg-size changes ($F_{(1,38)} = 0.05$–$2.8$, all $p>0.1$). Thus, *Rho1* knockdown consistently reduced egg size even after the experimental queens increased (Q7–Q12 after transfer into small colonies: $F_{(1,38)} = 45.6$–$654.8$, all $p<0.001$) or decreased (Q1–Q6 after transfer into large colonies: $F_{(1,38)} = 24.8$–$158.4$, all $p<0.001$) the egg size that they had produced in medium-sized colonies at the start of the experiment (**Figure 8B**, **Figure 8—source data 2**). All eggs appeared to be viable and were similar in color and texture when size-measured. However, their viability could not be confirmed because our size measurement is a destructive assay. Across individuals from all treatment groups, *Rho1* expression at the end of the experiment correlated almost perfectly with the produced egg size ($R_P = 0.98$, n = 12, $p<0.001$; **Supplementary file 6**). The correlation between *Rho1* expression and egg size was confirmed in a second dataset of 12 queens that produced small and large eggs due to different colony sizes ($R_P = 0.90$, n = 12, $p<0.001$; **Supplementary file 7**).

## Discussion

The egg is the major physical connection between generations and thus central to inter-generational epigenetic effects that have major implications for offspring phenotypes (*Krist, 2011*; *Tetreau et al., 2019*) and life history evolution (*Church et al., 2021*; *Rasanen and Kruuk, 2007*; *Church et al., 2019*; *Plaistow et al., 2006*). Despite its importance and its considerable inter- and intra-specific variability, the egg remains a poorly studied life history stage. Here, we provide evidence that egg size – a quantitative measure of maternal provisioning – is actively adjusted by honey bee queens in response to social cues that relate to colony size. We also find that queens in smaller colonies have smaller ovaries, presumably because they produce fewer but larger eggs. We find that protein localization, cytoskeleton organization, and energy generation are key proteomic changes in the ovary that mediate the production of large eggs. Finally, we identify the cytoskeleton organizer *Rho1* as a potentially important regulator of the active egg-size adjustment of honey bee queens.

Egg-size variation has been linked to parental or environmental conditions in numerous species (*Dani and Kodandaramaiah, 2017*; *Fox and Czesak, 2000*; *Sánchez-Tójar et al., 2020*), and we have previously provided evidence that honey bee queens also predictably adjust the size of produced eggs (*Amiri et al., 2020*). The direction of egg-size adjustments is consistent between solitary species and honey bees; egg size is typically increased under unpredictable or unfavorable conditions (*Einum and Fleming, 2004*; *Rollinson and Hutchings, 2013*) and positively correlated to maternal condition (*Fox and Czesak, 2000*; *Yu et al., 2022*). The first pattern holds in honey bees because egg size is decreased by the queen upon perception of a large colony. Larger eggs improve survival and offspring quality in honey bees (*Amiri et al., 2020*; *Yu et al., 2022*). Small colonies may select for increased individual survival because each individual is proportionally more significant to the colony (*Rueppell et al., 2009*), but large egg size in small colonies could also be directly related to less consistent brood care and overall colony-level resource availability. Conversely, smaller eggs in large colonies may be beneficial because invariably high levels of brood care do not require well-provisioned eggs. Although we do not know how feeding rates of queens change with colony size, maternal condition

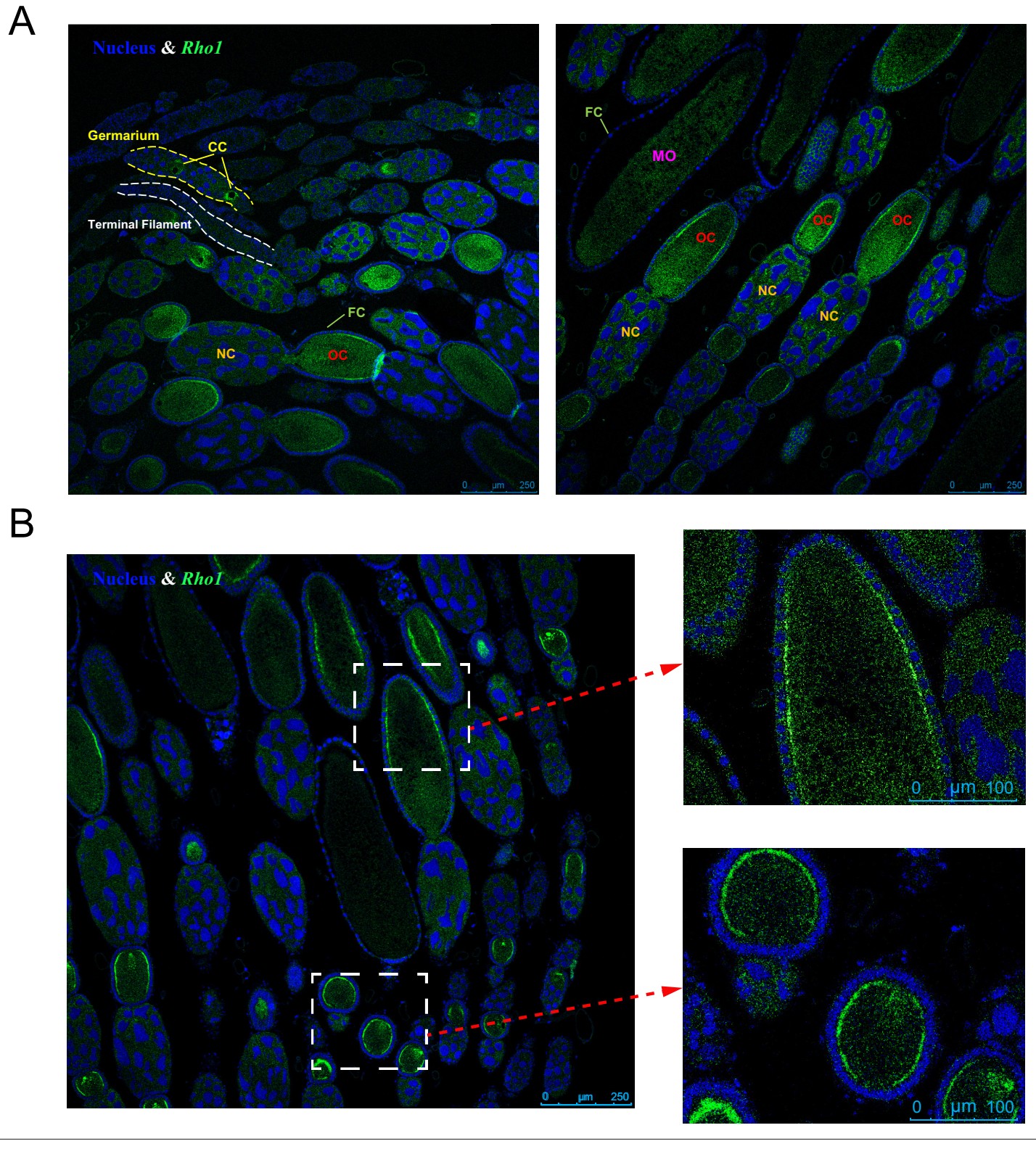

**Figure 7.** *Rho1* gene expression localization in the queen ovary via RNAscope in situ hybridization. (**A**) The expression of *Rho1* (green color) is limited to the growth stages of the oocyte: In the germarium, *Rho1* is expressed in cytocysts (CCs) and in the vitellarium, *Rho1* is highly expressed in oocytes (OCs), particularly near the lateral cell surface (**B**). In contrast, nurse cells (NCs) and follicle cells (FCs) do not exhibit elevated *Rho1* expression at this stage. Less expression of *Rho1* is observed in mature oocytes (MOs). Blue DAPI staining indicates cell nuclei for comparison. Representative pictures are shown, and the same patterns were found in six queens investigated.

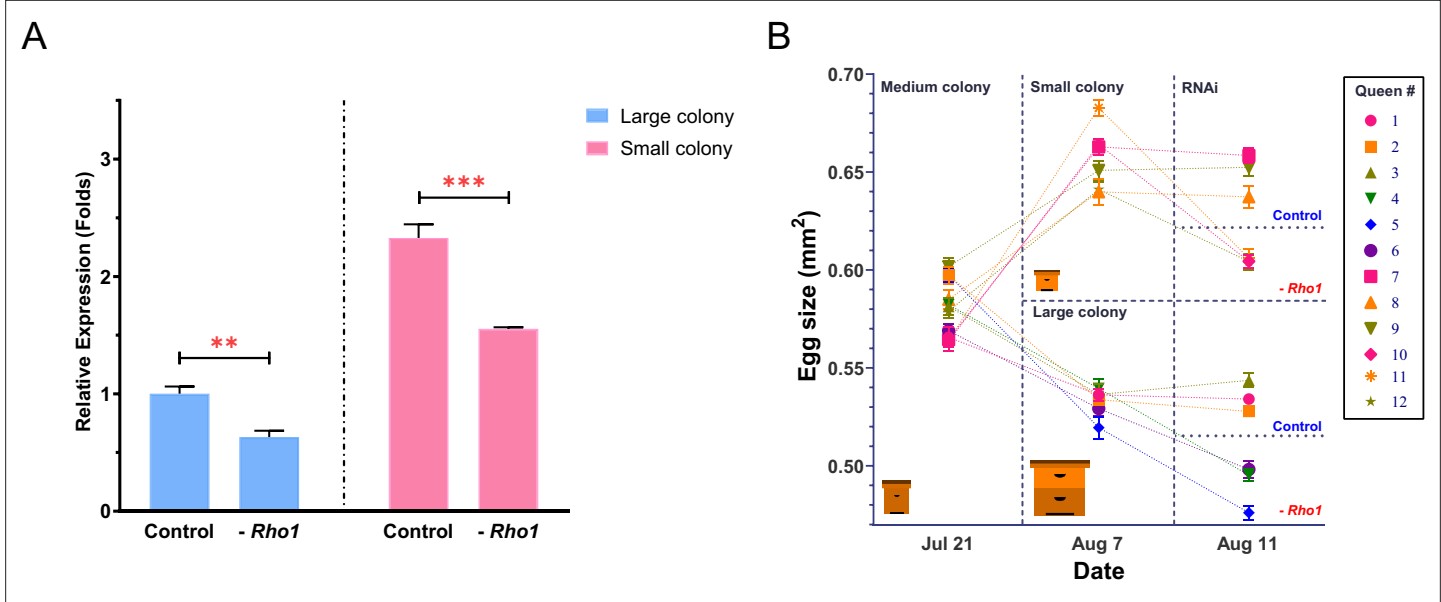

**Figure 8.** RNAi-mediated downregulation of *Rho1* decreases egg size in both 'small' and 'large' colonies. (**A**) RT-qPCR results confirmed the experimental downregulation of *Rho1* in ovaries of RNAi-injected queens and also showed that *Rho1* was significantly more expressed in queens that were housed in small colonies and thus produced larger eggs. Treatment effects were assessed by ANOVA with N = 6 and **p<0.01 and ***p<0.001. Means are shown with SD. (**B**) The 12 sister queens were mated and introduced to medium-sized colonies to establish egg laying. Subsequently, queens were randomly divided into two groups that were either introduced to small or large colonies. After the predicted egg-size differences were confirmed, three randomly chosen queens in each group were injected with *Rho1*-siRNA mix, and the other three were injected with scramble siRNA. Final egg-size measurements 3 days after injection demonstrated a significant reduction of egg size in all *Rho1* knockdown queens but not in control queens, regardless of colony environment. Means are shown with SEM.

The online version of this article includes the following source data for figure 8:

**Source data 1.** Gene expression data.

**Source data 2.** Egg-size measurements.

may also influence egg size in honey bees. For example, older queens produce smaller eggs than young queens (*Al-Lawati and Bienefeld, 2009*). However, our finding that queens in small colonies with reduced ovary size produce larger eggs suggests that an inverse relationship between individual condition and egg size can also exist in honey bees. Such a negative relationship between maternal condition and egg size may also exist in other social insects where colony-level instead of individual-level resource availability influences maternal resources and brood care performed by the workers (*Amdam and Omholt, 2002*).

The conventional trade-off between egg size and number (*Smith and Fretwell, 1974*; *Dani and Kodandaraiah, 2017*) may not apply to honey bees because resources can be distributed from other colony members to the queen (*Schrempf et al., 2017*; *Amdam and Omholt, 2002*; *Rueppell et al., 2016*). Accordingly, little evidence for a trade-off between egg size and number was found in a previous study of honey bee queens (*Amiri et al., 2020*). In contrast, a negative relation between egg size and number was found here, at least in the majority of comparisons between queens in small and large colonies. Queens in large colonies typically produce more eggs than queens in small colonies. Our finding that queens in large colonies have heavier ovaries indicates a physiological adaptation to satisfy the egg-laying demand in large colonies (*Al-Khafaji et al., 2009*). However, this result cannot explain why queens in smaller colonies produce larger eggs, why a temporary cessation of egg laying by queens in large colonies does not lead to an increase in egg size, and why queens decrease egg size upon their perception of being in a larger colony. Thus, a simultaneous regulation of egg size and egg number in response to perceived colony size is more likely than a direct trade-off between these two variables. However, further studies of resource flows to queens in colonies of different sizes are necessary to ultimately distinguish regulation from physiological or resource-based trade-offs. Quantitatively restricting egg-laying rates instead of preventing all egg laying is an additional experiment that can elucidate the relation between egg size, egg number, and colony size. The proposed active

regulation seems more likely and is also more compatible with short-term egg-size adjustments in other contexts (*Wei et al., 2019*) and can explain why queens in food-restricted colonies also produce larger eggs than queens with ample food supply (*Amiri et al., 2020*).

Our study further demonstrates that direct resource availability cannot explain the differences in egg size produced by honey bee queens in small versus large colonies. We find that connecting small colonies to another, larger colony without any physical contact leads to a reduction in egg size that is similar to the effect seen when queens are transferred between these colony types. Thus, our results suggest that the perception of colony size by the queen is sufficient for her to adjust the size of her eggs. We can exclude direct physical contact among individuals, which is used by worker honey bees to assess colony size (*Smith et al., 2017*), and thus we report a new modality by which honey bees perceive colony size. Multiple cues that travel through a double-screened tunnel could prompt the egg-size adjustment in queens, including sound, temperature, or pheromones and other semiochemicals. Future distinction among these possibilities will permit a subsequent investigation of the mechanisms by which social cues are translated into the physiological adjustments inside the ovary that we document here.

Broad comparisons find pronounced influences of social structure and behavior on egg size because variation in parental care is intricately linked to the initial investment in eggs (*Dixit et al., 2017*; *Summers et al., 2007*). Much less is known about social factors that lead to individual egg-size plasticity *Maeno et al., 2020*, particularly in cases that are as dynamic and reversible as illustrated here. We have not attempted to measure the speed at which queens adjust their egg size, but it could be much faster than the 1–2 weeks provided to queens in our experiments. Adjustments may even be made instantaneously, as queen- and worker-destined eggs differ in size (*Wei et al., 2019*) even though they are presumably produced at almost the same time. The advantageous caste bias of larger eggs (*Al-Kahtani and Bienefeld, 2021*) should select for paternal effects to increase egg size, in contrast to other polyandrous species in which males predominantly manipulate female fecundity (*Hollis et al., 2019*). Egg-size variation due to paternal manipulation remains to be investigated in honey bees in the context of the strong maternal control over egg size that we demonstrate in this study.

Due to the general paucity of a priori information on molecular mechanisms that determine egg size in insects (*Jha et al., 2015*), we used a naïve, quantitative proteomic comparison to identify the molecular causes of the egg-size plasticity in honey bee queens. The quantity of numerous proteins is associated with the production of either large or small eggs. Plod, which controls egg length in *Drosophila* (*Lerner et al., 2013*), is not among these proteins, but collagen IV, which may influence egg size in *Drosophila* indirectly (*Luo et al., 2021*), is found more abundantly in large-egg-producing ovaries. The vast majority (almost 95%) of differently abundant proteins are found at higher levels in ovaries that produce large eggs. Thus, the anatomically smaller ovaries are physiologically more active in several key processes than the larger ovaries that produce smaller eggs. The GO enrichment analysis indicates the prominence of two upregulated processes in large-egg-producing ovaries – 'protein localization' and 'cytoskeletal regulation' – while several energy metabolic processes are highlighted by the KEGG pathway analysis. These functional categories indicate that egg-size variation is not a simple increase of egg volume but reflects real differences in offspring provisioning, although the proteome of small and large eggs remains to be characterized (*McDonough-Goldstein et al., 2021*). Higher energy generation may be needed to produce more costly large eggs (*Wheeler, 1996*), and the cytoskeleton and protein localization processes are key to loading the eggs with nutrients in polytrophic ovaries (*Shimada et al., 2011*; *Wilson et al., 2011*). Several of the other GO terms, such as 'multicellular organism development' and 'oocyte construction,' are further plausible candidates to explain some of the observed variation in egg size.

Among all involved processes, we considered 'cytoskeletal organization' as the most likely regulatory mechanism, while other processes are more likely involved in downstream effector functions. Thus, we identified Rho1 as a potential candidate because it was centrally located in the PPI network of proteins related to cytoskeletal organization that were abundant in large-egg-producing ovaries. *Rho1* is a small, conserved GTPase with a likely role in egg-size regulation (*Murphy and Montell, 1996*) and thus also a plausible functional candidate. *Rho1* has multiple functions but generally plays an important role in cell morphogenesis by regulating the cytoskeleton (*Hall, 1998*). It primarily has been implicated in actin regulation (*Hall, 1998*), which is itself important for insect oogenesis (*Sokolova*

*et al., 2018*) but can also indirectly influence the microtubule network (*Pimm and Henty-Ridilla, 2021*). The regulation of *Rho1* activity is complex (*Denk-Lobnig and Martin, 2019*), and multiple participating signaling pathways could transduce extracellular signals in the ovary into cytoskeletal reorganization of the eggs.

The observed spatio-temporal expression of *Rho1* observed in our RNAscope experiment conforms well with the hypothesis that *Rho1* influences egg growth in the vitellarium. *Rho1* expression appeared to be focused in the cortex of oocytes close to the follicle cells, which mediate vitellogenin uptake (*Fleig, 1995*) and can influence egg size (*Wu et al., 2020*). Vitellogenesis is the period of rapid egg growth, and small differences during this time may cause significant differences in the size of the mature egg. However, the molecular function of *Rho1* remains to be elucidated. The role of *Rho1* in egg-size determination is further supported by the consistent decrease of egg size when *Rho1* is knocked down via siRNA injection. This specific effect occurs robustly in small- and large-egg-producing queens. Resulting eggs only differed in size, suggesting that the knockdown of *Rho1* does not cause pathological effects. The practical difficulties of genetic engineering in honey bees prohibited a complementary gain-of-function experiment. The almost perfect correlation between *Rho1* expression and egg size of queens across two different colony sizes and RNAi treatment groups strengthens the interpretation that social conditions, inter-individual differences, and RNAi manipulation all act through *Rho1* in a comparable manner to influence egg size. The tight correlation between egg size and *Rho1* expression was confirmed in a second data set.

Honey bee queens also adjust their egg size depending on whether a worker- or queen-destined egg is laid (*Wei et al., 2019*), and egg size influences the probability that an egg is raised into a future queen (*Al-Kahtani and Bienefeld, 2021*) and her reproductive quality (*Wei et al., 2019*; *Yu et al., 2022*). Our comparative results of the ovary proteome suggest that larger eggs are indeed of superior quality, but a comparison of the actual egg content remains to be performed to substantiate this argument. Stronger support than what we present here to tie egg-size regulation to *Rho1* may be difficult to obtain, but it can be tested whether *Rho1* relates to egg-size variation in honey bees in other contexts, such as maternal genotype (*Amiri et al., 2020*) or age (*Al-Lawati and Bienefeld, 2009*). If so, *Rho1* may become an important indicator of stress (*Amiri et al., 2020*) and maternal quality (*Yu et al., 2022*) in the ongoing efforts to sustain honey bee health. Comparative studies in models with a full suite of functional genomic tools, such as *Drosophila*, may be warranted to test the more general idea of *Rho1* as a key egg-size regulator in insects.

## Materials and methods

**Key resources table**

| Reagent type (species) or resource | Designation | Source or reference | Identifiers | Additional information |
|---|---|---|---|---|
| Sequence-based reagent | *Rho1*-siRNA | GenePharma | Cat# A01001 | See *Supplementary file 8* |
| Sequence-based reagent | Scramble siRNA | GenePharma | Cat# A06001 | See *Supplementary file 8* |
| Sequence-based reagent | PCR primers | Sangon Biotech | | See *Supplementary file 8* |
| Commercial assay or kit | PrimeScript RT reagent kit | TaKaRa | Cat# RR047A | |
| Commercial assay or kit | TB Green Fast qPCR Mix | TaKaRa | Cat# RR430A | |
| Commercial assay or kit | RNAscope Multiplex Fluorescent Reagent Kit v2 | Advanced Cell Diagnostics | Cat# 323100 | |
| Commercial assay or kit | RNAscope Probe- Amel-LOC409910-C1 | Advanced Cell Diagnostics | Cat# 1061331-C1 | |
| Chemical compound, drug | DiI stain | Beyotime | Cat# C1036 | |
| Software, algorithm | ImageJ | https://Imagej.nih.gov/ij | | |
| Software, algorithm | GraphPad Prism v8 | GraphPad Software | | |

*Continued on next page*

*Continued*

| Reagent type (species) or resource | Designation | Source or reference | Identifiers | Additional information |
|---|---|---|---|---|
| Software, algorithm | SPSS Statistics 20.0 | IBM | RRID:SCR_019096 | |
| Software, algorithm | Xcalibur 3.0 | Thermo Fisher Scientific | RRID:SCR_014593 | |
| Software, algorithm | PEAKS 8.5 | Bioinformatics Solutions | | |
| Software, algorithm | Primer Premier 5.0 | PREMIER Biosoft | | |
| Software, algorithm | Cytoscape v3.8.2 | https://cytoscape.org | | |
| Software, algorithm | TBtools | https://doi.org/10.1016/j.molp.2020.06.009 | | |
| Software, algorithm | STRING | https://string-db.org | | |
| Software, algorithm | Metascape | http://metascape.org/ | | |
| Software, algorithm | geNorm | https://genorm.cmgg.be | | |

## Experimental model and subject details

All studies were conducted in the Western honey bee, *A. mellifera*, using colonies of mixed origin and derived from commercial populations, which were kept in the research apiary of the University of North Carolina at Greensboro, NC, USA (UNCG: 2020) or in the research apiary of the Institute of Apicultural Research in Beijing, China (IAR: 2021). We used standard husbandry methods to house experimental colonies (*Laidlaw and Page, 1997*), monitoring and adjusting colony size and food status but refraining from any other treatments during the experiments. We defined three distinct colony sizes: 'small' colonies contained 500–700 worker bees and housed in mating hives (nucs) equipped with three half-frames of medium depth, 'medium' colonies contained 6000–8000 workers bees housed in a 5-frame Langstroth hive box with standard frames, and 'large' colonies with 16,000–20,000 worker bees in a standard 8-frame (UNCG) and 10-frame (IAR) Langstroth hive. Thus, all colony sizes were below the size of apicultural production colonies. Each separate experiment was conducted with a set of sister queens that we raised from offspring of a single mother to reduce genetic variation within experiments. Queen rearing followed standard methods (*Laidlaw and Page, 1997*), and queens were allowed to mate naturally.

## Repeated transfer experiments

As an extension of our previous study (*Amiri et al., 2020*), an experiment was set up in the UNCG apiary to transfer one group of queens from 'medium' to 'small' to 'large' colonies and simultaneously transfer another group from 'medium' to 'large' to 'small' colonies. During each stage, egg size was measured from 20 eggs per queen twice (1 week apart). The measurements followed our previous protocol (*Amiri et al., 2020*), where eggs produced overnight were randomly selected in the next morning and transferred with a grafting tool from standard worker brood cells onto a 0.01 mm stage micrometer (Olympus, Japan). Eggs were laterally photographed under threefold magnification while ensuring that the egg was completely level. Each photo was then processed with the open-source ImageJ software (version 1.52p; National Institutes of Health, USA) by manually tracing the egg's outline using the polygon-selection tool. The selected area was measured in mm$^2$ (note that in our previous work [*Amiri et al., 2020*], a simple conversion mistake led to erroneous µm$^2$ units) as a representation of egg size. This two-dimensional measure of egg size directly corresponds to egg mass, assuming a cylindrical egg shape, and is more precise than weight determination on any scale available to us. During our measures of hundreds of eggs, we did not observe any obvious deviations from a cylindrical egg shape, but any such deviations would make our measures imprecise.

From an original 16 queens, 11 were successfully mated and started egg laying in their respective 'medium' colonies. After 2 weeks and two egg-size measurements (on June 12 and 19, 2020), six of these queens were transferred to 'small' colonies, while the remaining five were transferred to 'large' colonies. After 2 weeks of acclimation, another two egg-size measurements from each queen were performed (on July 7 and 14, 2020, = week 3 and week 4) before reciprocally transferring queens between 'small' and 'large' colonies. This transfer was survived by five queens ending up in 'small'

colonies and three queens in 'large' colonies. After another 2 weeks of acclimation, egg size of these remaining queens was measured as before (on August 4 and 11, 2020, = week 5 and week 6). In addition to egg size, the number of eggs laid over 15 hr by queens was determined.

Egg-size differences among queens at the beginning of the experiment were tested by simple ANOVA across all queens. Overall changes in average egg size and number between the first and second week of the experiment were assessed by a paired *t*-tests, which also assessed individual consistency by correlation. Subsequent egg-size changes were assessed separately for each queen between sample points by simple ANOVA. The relation between average egg size and egg number was assessed by Pearson's correlation at each sample point separately. All statistical tests were performed in IBM's SPSS 28.0, and significance values were determined by bootstrapping.

At the conclusion of this experiment, all surviving queens were weighed and sacrificed for determination of their body size and ovary weight (*Figure 2A*) and subsequent analyses of the ovary proteome. To increase sample size, two additional queens were transferred from 'medium' colonies and housed in 'small' colonies for 2 weeks before including them in these analyses. Queens were captured alive and weighed in a pre-weighed 1.5 ml tube centrifuge tube to the nearest microgram. We determine fresh weight because this measure is often used in practice even though it varies on a short-term basis with feeding, defecation, and oviposition. After cold anesthetization, both forewings were detached from the queens and mounted on a microscope slide to determine the distance between the distal end of the marginal cell and the intersection of the Cu1 and 2m-cu veins as a representative size measure (*Waddington, 1989*). The values from both wings were averaged. This measure provides a well-defined and highly repeatable measure of wing size, which integrates length and width and is largely unaffected by wing wear. Subsequently, the ovary was dissected from the chilled abdomen, weighed, and frozen at –80°C for proteome profiling (see below). Great care was taken to avoid contamination with other tissues, and we cannot exclude the possibility that small parts of the ovary were discarded with the surrounding tissue, although the relatively large size of the honey bee queen ovary makes gross mistakes in determining ovary weight due to such errors negligible.

A simpler, additional experiment was conducted to perform another comparison of queen- and ovary weight between queens in 'small' and 'large' colonies in order to gather more ovaries from these treatment groups for proteome profiling. Accordingly, sister queens were reared from a randomly selected mother in the UNCG apiary. After their maturation into ovipositing queens in 'medium' colonies, four were successfully introduced into 'small' colonies and four others into 'large' colonies. After 2 weeks, the production of large and small eggs respectively was confirmed for all eight accepted queens and queens were compared with regard to their body weight, body size, and ovary weight (*Figure 2B*). The ovaries of these queen were also collected and stored at –80°C for proteome profiling.

A third study of ovary size was performed with all queens at the end of the RNAi knockdown experiment (see below). For this purpose, the body weight, wing size, and ovary weight of the 12 queens were measured as described above (*Figure 2C*). In this instance, the body weight was measured before injection, while the wing size and ovary weight were measured after injection.

## Oviposition restriction experiment

Prompted by the equivocal evidence for a negative correlation between egg size and egg-laying rate in the first experiment and the lack of such a correlation in our previous study (*Amiri et al., 2020*), we explicitly tested the hypothesis that egg size is an invariant consequence of the egg-laying rate of honey bee queens. Eight sister queens were reared from a randomly selected source in the IAR apiary in July 2021, introduced as mature queen cells to 'medium' colonies for emergence, mating, and initiation of oviposition. Subsequently, the queens were transferred into 'large' colonies, and 2 weeks after acceptance their egg sizes were measured as described above. Subsequently, queens were randomly split into an oviposition restriction group and an unmanipulated control group. Queens in the oviposition restriction group were caged in push-in queen cages (36 cm × 18 cm) on top of capped brood combs without empty cells as egg-laying opportunities. Thus, queens in this treatment group experienced normal colony conditions without an opportunity to lay eggs, while the control queens were left unmanipulated. Immediately after these 2 weeks, all queens were caged on identical sections of comb with empty cells to measure their egg sizes again. Each queen was evaluated

separately for significant differences in egg size between the start and end of the experiment with a simple ANOVA.

## Colony extension experiment

To clarify how colony size influences queen oviposition, we tested whether physical contact or material transfers are necessary to alter the size of eggs produced by the queen. Six sister queens were reared from a randomly selected mother in the IAR apiary in July 2021. After maturation (as described above), these queens were introduced into 'small' colonies. After an acclimation period of 2 weeks, egg sizes produced by all queens were determined twice as described above (15th and 21st of August 2021). The colonies were randomly divided into two groups and connected via a double-screened tunnel to a 'medium' hive that contained either empty comb (control) or a 'mMedium' colony with corresponding amounts of food, brood, and workers, but no queen (treatment). Tunnels were 3 cm long, 10 cm wide, and 20 cm high. Both ends were screened with fine wire mesh to prevent any physical contact among the workers in opposing hives. Worker drifting between hives was prevented by pointing the hive entrances of the two connected units to opposite directions, as well as coloring and designing the entrances differently. One week later, egg-size measurements were performed (30th of August) and repeated once after an additional week (4th of September).

## Ovary proteome analysis

In an unbiased search for differences, the protein content of ovaries that produce small eggs (from queens in 'large' colonies) and ovaries that produce large eggs (from queens in 'small' colonies) was studied with a label-free LC-MS/MS approach. A total of 18 ovaries, collected during the two UNCG experiments described above, were included. For both groups (small-egg-producing queens and large-egg-producing queens), nine ovaries were pooled randomly into three biological replicates.

Total protein was extracted using previously described methods (*Fang et al., 2014*). Protein concentration was determined using a Bradford assay, and the general quality of extracted proteins was confirmed by SDS-PAGE with Coomassie Blue staining. An aliquot of 200 μg of protein from each pool was reduced with DTT (final concentration 10 mM) for 1 hr, then alkalized with iodoacetamide (final concentration 50 mM) for 1 hr in the dark. Thereafter, protein samples were digested at 37°C overnight with sequencing-grade trypsin (enzyme: protein (w/w) = 1:50). The digestion was stopped by adding 1 μl of formic acid then desalted using C18 columns (Thermo Fisher Scientific, USA). The desalted peptide samples were dried and dissolved in 0.1% formic acid in distilled water, then quantified using a NanoDrop 2000 spectrophotometer (Thermo Fisher Scientific) and stored at –80°C for subsequent LC-MS/MS analysis.

LC-MS/MS analysis was performed on an Easy-nLC 1200 (Thermo Fisher Scientific) coupled Q-Exactive HF mass spectrometer (Thermo Fisher Scientific). Buffer A (0.1% formic acid/water) and buffer B (0.1% formic acid and 80% acetonitrile in water) were used as mobile phase buffers. Peptides were separated using a reversed-phase trap column (2 cm long, 100 μm inner diameter, filled with 5.0 μm Aqua C18 beads; Thermo Fisher Scientific) and an analytical column (15 cm long, 75 μm inner diameter, filled with 3 μm Aqua C18 beads; Thermo Fisher Scientific) at a flow rate of 350 nl/min with the following 120 min gradients: from 3 to 8% buffer B in 5 min, from 8 to 20% buffer B in 80 min, from 20 to 30% buffer B in 20 min, from 30 to 90% buffer B in 5 min, and remaining at 90% buffer B for 10 min. The eluted peptides were injected into the mass spectrometer via a nano-ESI source (Thermo Fisher Scientific). Ion signals were collected in a data-dependent mode and run with the following settings: scan range: m/z 300–1800; full scan resolution: 70,000; AGC target: 3E6; MIT: 20 ms. For MS/MS mode, the following settings were used. Scan resolution: 17,500; AGC target: 1E5; MIT: 60 ms; isolation window: 2 m/z; normalized collision energy: 27; loop count 10; dynamic exclusion: 30 s; dynamic exclusion with a repeated count: 1; charge exclusion: unassigned, 1, 8, >8; peptide match: preferred; exclude isotopes: on. The corresponding raw data were retrieved using Xcalibur 3.0 software (Thermo Fisher Scientific).

The extracted MS/MS spectra were searched against the protein database of *A. mellifera* (23,430 sequences, from NCBI) appended with the common repository of adventitious proteins (cRAP, 115 sequences, from The Global Proteome Machine Organization) using PEAKS 8.5 software (Bioinformatics Solutions, Canada). The search parameters were ion mass tolerance, 20.0 ppm using monoisotopic mass; fragment ion mass tolerance, 0.05 Da; enzyme, trypsin; allow nonspecific cleavage at none

end of the peptide; maximum missed cleavages per peptide, 2; fixed modification, carbamidomethylation (C, +57.02); variable modifications, oxidation (M, +15.99); maximum allowed variable PTM per peptide, 3. A fusion target-decoy approach was used for the estimation of false discovery rate (FDR) and controlled at ≤1.0% (−10 log p≥20.0) both at peptide and protein levels. Proteins were identified based on at least one unique peptide.

Quantitative comparison of the egg proteome between the two experimental groups was performed by the label-free approach embedded in PEAKS Q module. Feature detection was performed separately on each sample by using the expectation-maximization algorithm. The features of the same peptide from different samples were reliably aligned together using a high-performance retention time alignment algorithm. Significance was calculated by ANOVA, using a threshold of p≤0.01. Results were visualized as a heatmap using TBtools software (*Chen et al., 2020*), clustering based on Euclidean distance and the 'complete' method. The LC−MS/MS data and search results were deposited in ProteomeXchange Consortium (http://proteomecentral.proteomexchange.org) via the iProX partner repository with the dataset identifier PXD029859.

For further functional analysis, honey bee proteins were mapped to their *Drosophila melanogaster* homologs using KOBAS 3.0 (*Xie et al., 2011*). Proteins of interest were uploaded as fasta sequences, *D. melanogaster* was selected as target species, and similarity mapping was conducted with default cutoffs (BLAST *E*-value < 1E−5 and rank ≤ 5).

Functional GO enrichment analysis of the quantitatively different proteins was performed based on biological processes with ClueGO+ CluePedia version 2.5.7 (*Bindea et al., 2009*), a Cytoscape (version 3.8.2) plugin. Two-sided hypergeometric test (enrichment/depletion) with p-value≤0.05 was used followed by Bonferroni correction. The GO tree interval was set between 3 and 8, with minimum of five genes and 1% of genes. Kappa score ≥ 0.4 was applied to generate term–term interrelations and functional groups based on shared genes between the terms. KEGG pathway enrichment analysis was done in Metascape (http://metascape.org/) with default settings: minimum overlap, 3; p-value cutoff, 0.01; minimum enrichment, 1.5.

For the exploration of functional protein connections involved in the major enriched biological process terms, PPI networks were constructed among the differing proteins in STRING (*Szklarczyk et al., 2015*). A full STRING network was built with medium confidence (0.4) and FDR < 5%. The PPI networks were visualized using Cytoscape (version 3.8.2).

## Examination of expression patterns of *Rho1*

Based on the proteomic analyses, the small GTPase *Rho1* emerged as a candidate regulator of egg size during honey bee oogenesis, which motivated us to study its expression patterns in the ovary and inside the oocyte by RNAscope in situ hybridization (*Wang et al., 2012*). The probes were designed and prepared by Advanced Cell Diagnostics (ACD, Inc, Hayward, USA), and an RNAscope Fluorescent Multiplex Reagent kit (ACD) was used following the manufacturer's instructions. Immediately following dissection, the ovary tissues of randomly selected, mature queens from the IAR apiary were fixed in 10% neutral buffered formalin for 32 hr at room temperature (RT). Thereafter, the samples were dehydrated using a standard ethanol series, followed by xylene. The dehydrated samples were embedded in paraffin and then cut into 1 μm sections using an RM2235 microtome (Leica, Germany) that were gently deposited onto glass microscope slides. The slides were then baked for 1 hr at 60°C and deparaffinized at RT. The sections were treated with hydrogen peroxide for 10 min at RT and then washed with fresh distilled water.

Subsequently, the target retrieval step was performed using 1× RNAscope target retrieval reagent. The slides were air-dried briefly and then boundaries were drawn around each section using a hydrophobic pen (ImmEdge pen; Vector Laboratories, USA). After hydrophobic boundaries had dried, the sections were incubated in protease IV reagent for 2 min, followed by a 1× PBS wash. Each slide was then placed in a prewarmed humidity control tray (ACD) containing dampened filter paper and incubated in a mixture of Channel 1 probes (*Rho1*, ACD Cat #1061331-C1) for 2 hr in the HybEZ oven (ACD) at 40°C. Following probe incubation, the slides were washed two times in 1× RNAscope wash buffer and returned to the oven for 30 min after submersion in AMP-1 reagent. Washes and hybridization were repeated using AMP-2, AMP-3, and HRP-C1 reagents with a 30 min, 15 min, and 15 min incubation period, respectively. The slides were then submerged in TSA Plus FITC and returned to the oven for 30 min. After washing two times in 1× RNAscope wash buffer, the slides were incubated with

HRP blocker for 15 min in the oven at 40°C. Finally, the slides were washed two times in 1× RNAscope wash buffer and incubated with DAPI for 1 min. The images were visualized with a Leica SP8 (Leica) confocal microscope and acquired with the sequence program of the Leica LAS X software.

## RNAi-mediated downregulation of *Rho1*

To test the hypothesis that *Rho1* expression controls the size of the eggs that honey bee queens produce, we investigated the effects of RNAi-mediated downregulation of *Rho1*. Four specific siRNAs targeting Rho1 of *A. mellifera* (GenBank: LOC409910) were designed and synthesized by GenePharma RNAi Company (Shanghai, China). Scrambled siRNA of random sequence was used as a negative control (GenePharma). For all siRNA sequences, see *Supplementary file 8*.

Twelve sister queens were produced from a random source hives of the IAR apiary. They were introduced into 'medium' colonies to mate and establish egg laying. When a regular laying pattern was established, the size of the eggs produced by all queens was measured as described above. Then, queens were randomly divided into two groups of six that were either introduced into 'small' or 'large' colonies. Two weeks after queen acceptance, egg-size measurements were repeated. One day later, three queens in each group were randomly selected and injected with 1 µl/queen of *Rho1*-siRNA mix (mixture of the four *Rho1*-siRNAs, 1 µg/µl), and the other three were injected with 1 µl/queen of scrambled siRNA (1 µg/µl). The queens were transferred to the laboratory and narcotized with $CO_2$ before injection. Injections were made dorsally between the fourth and fifth abdominal segment of queens using a microliter syringe (NanoFil; World Precision Instruments, USA) coupled with a 35G needle (NF35BV-2; World Precision Instruments). Injected queens were given time to recover and placed back into their original colonies. Egg-size measurements for each queen were performed 3 days after injection, and control or treatment effects on egg size were tested separately for each queen by comparing the sizes produced before and after RNAi injection.

To assess the efficacy of RNAi knockdown of *Rho1* and investigate the correlation between ovary size and *Rho1* expression, the expression of *Rho1* was quantified by quantitative real-time PCR. Queens were anesthetized before dissection of the ovary for weight measurement (see above) and subsequent RNA extraction, cDNA synthesis, and RT-qPCR according to previously described methods (*Han et al., 2021*). The average of three technical replicates were computed and used in subsequent analyses. Reference genes were evaluated by GeNorm analysis (*Vandesompele et al., 2002*) that indicated all evaluated reference genes (*Arfgap3, CylD, GAPDH, Keap1, Kto, mRPL44, RpA-70, Rpn2*) had high expression stability expression across samples (M-values < 0.4). Following GeNorm's recommendation, we used *Arfgap3* and *CylD* to calculate relative gene expression as $2^{-\Delta\Delta Ct}$ (*Livak and Schmittgen, 2001*). A corresponding analysis to confirm the correlation between *Rho1* and egg size was performed in a second set of 12 queens without RNAi exposure.

## Acknowledgements

We thank all our lab members for their encouragement and support. This study was funded by grants from the National Natural Science Foundation of China (31970428) and the China Scholarship Council (201903250009) to BH, a scholarship to EA by the US National Research Council, a grant to SX by the Agricultural Science and Technology Innovation Program (CAAS-ASTIP-2015-IAR), a grant to JL by the earmarked fund for Modern Agro-Industry Technology Research System (CARS-44) in China, and grants from the US Army Research Office (W911NF1920161 and W911NF2210195), the Natural Sciences and Engineering Research Council of Canada (RGPIN-2022-03629), and the Alberta Beekeepers Commission to OR.

## Additional information

### Funding

| Funder | Grant reference number | Author |
| --- | --- | --- |
| National Natural Science Foundation of China | 31970428 | Bin Han |

| Funder | Grant reference number | Author |
| --- | --- | --- |
| China Scholarship Council | 201903250009 | Bin Han |
| National Research Council | Postdoctoral Fellowship | Esmaeil Amiri |
| Agricultural Science and Technology Innovation Program | CAAS-ASTIP-2015-IAR | Shufa Xu |
| Earmarked Fund for Modern Agro-industry Technology Research System | CARS-44 | Jianke Li |
| Army Research Office | W911NF1920161 | Olav Rueppell |
| Army Research Office | W911NF2210195 | Olav Rueppell |
| Natural Sciences and Engineering Research Council of Canada | RGPIN-2022-03629 | Olav Rueppell |
| Alberta Beekeepers Commission | | Olav Rueppell |

The funders had no role in study design, data collection and interpretation, or the decision to submit the work for publication.

## Author contributions

Bin Han, Conceptualization, Formal analysis, Supervision, Funding acquisition, Investigation, Writing - original draft, Writing - review and editing; Qiaohong Wei, Han Hu, Lifeng Meng, Investigation; Esmaeil Amiri, Conceptualization, Formal analysis, Investigation, Writing - original draft, Writing - review and editing; Micheline K Strand, Supervision, Funding acquisition, Writing - review and editing; David R Tarpy, Supervision, Writing - review and editing; Shufa Xu, Conceptualization, Resources, Funding acquisition, Writing - review and editing; Jianke Li, Resources, Funding acquisition, Writing - review and editing; Olav Rueppell, Conceptualization, Resources, Formal analysis, Supervision, Funding acquisition, Writing - original draft, Writing - review and editing

## Author ORCIDs

Bin Han ![ORCID] http://orcid.org/0000-0001-6974-8699
David R Tarpy ![ORCID] http://orcid.org/0000-0001-8601-6094
Jianke Li ![ORCID] http://orcid.org/0000-0002-9344-0886
Olav Rueppell ![ORCID] http://orcid.org/0000-0001-5370-4229

## Decision letter and Author response

Decision letter https://doi.org/10.7554/eLife.80499.sa1
Author response https://doi.org/10.7554/eLife.80499.sa2

# Additional files

## Supplementary files

- Supplementary file 1. Egg-size measurements from queen ovary comparison experiments.
- Supplementary file 2. Proteins of difference abundance.
- Supplementary file 3. Gene Ontology results of upregulated proteins in large-egg-producing ovaries.
- Supplementary file 4. KEGG pathway results of upregulated proteins in large-egg-producing ovaries.
- Supplementary file 5. Protein–protein interaction analysis of cytoskeleton organization proteins.
- Supplementary file 6. *Rho1* expression and egg-size correlation analysis from the RNAi experiment.
- Supplementary file 7. *Rho1* expression and egg-size correlation analysis using an independent dataset.

- Supplementary file 8. Sequence information for RNAi and qPCR.
- MDAR checklist

## Data availability

The LC-MS/MS data and search results were deposited in ProteomeXchange Consortium (http://proteomecentral.proteomexchange.org) under accession number PXD029859. All other data are provided as supplementary files.

The following dataset was generated:

| Author(s) | Year | Dataset title | Dataset URL | Database and Identifier |
|---|---|---|---|---|
| Han B, Xu S | 2022 | Honeybee queen ovary proteomics | http://proteomecentral.proteomexchange.org/cgi/GetDataset?ID=PXD029859 | ProteomeXchange, PXD029859 |

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
