## [Editor Report]

This study provides valuable insights into the control of egg size plasticity, a key form of maternal investment. It presents convincing evidence from both experimental manipulations and molecular investigations of egg plasticity in honey bee queens. It will be of interest to evolutionary biologists, particularly those working on life-history trade-offs and reproductive strategies.

---

## [Decision Letter]

**Decision letter after peer review:**

Thank you for submitting your article "Social Regulation of Egg Size Plasticity in the Honey Bee is Mediated by Cytoskeleton Organizer Rho1" for consideration by *eLife*. Your article has been reviewed by 3 peer reviewers, one of whom is a member of our Board of Reviewing Editorss, and the evaluation has been overseen by K VijayRaghavan as the Senior Editor. The following individual involved in review of your submission has agreed to reveal their identity: Seth Donoughe (Reviewer #2).

The reviewers have discussed their reviews with one another, and the Reviewing Editor has drafted the comments below to help you prepare a revised submission if you choose to do resubmit your work to *eLife*. Overall the reviewers agreed that your study was interesting, but that extensive revisions and/or additional data are needed to justify the current conclusions.

Essential revisions:

1) Revise and qualify the conclusions on the role of Rho1 as a key regulator for egg size, in line with comments from Reviewers 2 and 3. Specifically, there is a need to better justify the idea that Rho1 is a "social mediator" of egg size as proposed in the title.

2) Revise the section describing an effort to investigate the effects of egg-laying rate on egg size. To test this, you would need to compare queens in which egg-laying rate was experimentally increased or decreased to those where it was not. The presented manipulation instead compares queens in which egg-laying was ceased completely. If more appropriate data for this key question can be added please do so. Or the conclusions must be revised to better reflect the actual manipulation.

3) A table or other system to allow readers to better keep track of the experimental manipulations of each queen and their use in the proteomic analyses (see Reviewer 2 comment 2).

4) Greater clarity on the relationship between body size and egg size. You may follow the recommendation of Reviewer 2 (Comment 3): "Please add a scatter plot of wing size vs. baseline ovary size and wing size vs. baseline egg size (one point per queen). This will help your readership to better understand what is giving rise to the individual-level and species-level variation in egg and ovary size. Also, please add a few sentences to the Results to explain why you selected these traits to measure from each queen: that particular landmark to landmark distance on the wings, ovary weight, and this method of egg size measurement; also please include a sentence or two to summarize any important limitations of these selected measures."

5) More data or justification on the fate of larger vs smaller eggs in honey bees, so that the likely ecological and evolutionary significance of the reported changes in egg size is clear.

*Reviewer #1 (Recommendations for the authors):*

This study makes several new contributions to our understanding of maternal control of egg size in honey bees. It seems reasonable to assume that larger eggs compensate in some way for smaller colonies (e.g. offset in some way the better provisioning given to newly-hatched larva in larger colonies?) but the actual significance of egg size is not much discussed in this study. I was left wondering whether larger worker eggs have been previously established to have higher survival in honey bees (if so, citations of those studies are needed) or whether this remains an assumed outcome of differences in egg size. Either way, I recommend more in the Intro or Discussion reflecting on the possible importance of egg size for fitness in the context of the honey bee life cycle.

Other comments:

80: "..an adaptation to the colonial life cycle of honey bees but maybe more generally applicable to other oviparous animals." How might social cues affect oviparous animals that do not live in social groups?

86: Cite your previous study here when you refer to it.

88: Give N here and elsewhere in Results when treatments are first introduced (several times I had to scroll down to Methods in order to find information about sample sizes).

119-126: I appreciate the effort here to experimentally investigate whether egg size is affected by egg-laying rate, but the question does not seem to be entirely answered. In this experiment, queens from large colonies are restricted from any egg-laying and then have their eggs measured again once egg-laying resumes. This is not really the same as simulating ongoing lower egg-laying rate (as might be experienced by a queen in a small colony). Your conclusion from this section is that egg size is not a by-product of higher or lower egg number, but is that saying that the two (number and size) are unrelated? Overall, I think the question of whether this important trade-off exists in honey bees warrants more discussion.

187: "All eggs appeared to be viable and differed phenotypically only in size" – was the viability of at least some eggs confirmed?

*Reviewer #2 (Recommendations for the authors):*

(1) Explanations for general readers. Given the broad readership of this journal, and the wide interest that there will be for this topic and your findings, we ask that you make the Results more approachable to readers who are not familiar with culturing honeybees. For instance at the beginning of the Results section, please give a general explanation of how these experiments are done. Some questions that a non-specialist reader will have: Can queens be moved between colonies that they did not found? When a queen was moved into a new colony, did the previous queen need to be removed first? What are sister queens, and is it important that sister queens be used for this sort of experiment? Also please give a few sentences of explanation to contextualize the categories of small, medium, and large colonies. How do these sizes compare to average/healthy colony sizes? Does the number of queens in a colony increase with colony size, is it controlled here?

(2) Clarity in how you refer to treatments. I had some difficulty keeping track of the different treatments. Please add a table in which each row represents a distinct treatment. Make a unique text label for each unique treatment, and then give the relevant treatment details (E.g. "R1" : "Queen spent A days in colony of B size, then spent C days in colony of D size, then spent E days in colony of F size"). Then in the text you can use the labels to refer readers to the exact treatments ("We did proteomic analysis on queens from groups R1 and R2"), and then you can use more colloquial language to describe the results ("The ovaries that made large eggs were enriched with…"). Once this has been done, use these labels to make the figures clearer. Namely, in a given figure, all points that were from the same treatment group should be given the same color, and each individual queen from that treatment group should have a distinct symbol shape.

(3) Wing size, ovary size, and interindividual variation. The data that have been collected-wing size, ovary size, and egg size for all the queens-can help to shed light on sources of egg size variation. Wing size cannot change during adulthood and so it can be treated as a measure of overall size/nutrition during initial queen maturation, while ovary size and egg size are the factors that respond to the environmental condition. Please add a scatter plot of wing size vs. baseline ovary size and wing size vs. baseline egg size (one point per queen). This will help your readership to better understand what is giving rise to the individual-level and species-level variation in egg and ovary size. Also, please add a few sentences to the Results to explain why you selected these traits to measure from each queen: that particular landmark to landmark distance on the wings, ovary weight, and this method of egg size measurement; also please include a sentence or two to summarize any important limitations of these selected measures.

(4) How does the shape of eggs change when egg size changes?

(5) Do you have data on the number of ovarioles for any of the queens? If so, please add a plot showing the relationship between ovariole number and baseline egg size.

(6) This is a more detailed explanation for a comment I made in the Public Review. The experiment that is described as studying "egg laying rate" does not really address egg laying rate. As I understand it, the queens are moved to a place where there are no suitable places to lay eggs. It seems that the authors are saying that even if the queen can't lay eggs, the eggs don't just get bigger and bigger in the ovaries while they wait, and this is evidence that there isn't a relationship between laying rate and size. Based on the existing literature, I don't think there is any reason to think that restricting oviposition would or could cause an increase in egg size. In general, when insects don't have a good oviposition site, they pause oogenesis, resorb the eggs, or lay the eggs anyway (in a non-suitable oviposition site). The cleaner way to ask if there is a relationship between egg size and laying rate is to start by quantifying it in your size-manipulation experiments. Why not just count how many eggs the queens are laying per [unit of time] in the big and small colonies? If you do not have the data for that, that's fine, but that is certainly what researchers generally do when studying egg laying rate. Bottom line: this experiment does answer some other interesting questions, but not about egg-laying rate. Please re-frame this section in terms of "restricted oviposition" or some other phrase that you prefer.

(7) Please be sure sample sizes are included throughout the text and in figure legends (egg measurements and individual queens), and in the new table recommended in Major Point (2).

(8) Include some discussion of the stages of oogenesis in the Results and/or Discussion. When an adult is increasing or decreasing the size of the eggs she lays, this is most likely happening by changing the amount of time that the developing eggs spend in one or more stages of oogenesis (and the leading candidate would be the period of vitellogenesis, presumably). This is also the simplest explanation for the differences in protein expression levels. In the sections on proteomics, gene expression patterns, and/or RNAi, consider if any discussion of this topic might be informative and if so include it. It is probably beyond the scope of this project to carefully analyze the process of oogenesis that gives rise to the various sizes, but that would be a very interesting future direction. On a related point, I am quite confused by the ovary weight finding. How does the ovary weigh less with bigger eggs? Are there fewer eggs? That is what I assumed but I could be wrong. Please include a bit more discussion of this surprising finding, in the Results or Discussion.

(9) I am excited about the authors' idea that this is life history without the nutrition limitations, but it will be helpful to give readers a little information about queen nutrition (in the Results and/or Discussion). How would her amount of food differ between small and large colonies? Do 10x more workers means she eats 10x as much?

(10) A few relevant sources to cite. There is another instance of plastic changes to eggs during insect oogenesis in response to perception: Abram, Paul K., et al. "An insect with selective control of egg coloration." Current Biology 25.15 (2015): 2007-2011. Also there was a recent effort to determine if egg size is related to egg number (or other things), and it does include social insects (see lines 41-42): Church, Samuel H., et al. "Repeated loss of variation in insect ovary morphology highlights the role of development in life-history evolution." Proceedings of the Royal Society B 288.1950 (2021): 20210150.

*Reviewer #3 (Recommendations for the authors):*

Two major conclusions are drawn in the manuscript by Han et al.: (1) that queens predictably and reversibly increase egg size in small colonies and decrease egg size in large colonies, while their ovary size changes in the opposite direction; and (2) that Rho1 is a candidate regulator of egg size. The Authors interpret these two findings in the following way (Lines 201 and 202): "we identify the cytoskeleton organizer Rho1 as a key regulator of the active egg size adjustment of honey bee queens." While I admire the work that Authors have done here, I have the following major concerns:

Major Concern # 1: The authors knock down Rho1 using RNAi and find that it reduces egg size in both large and small colony sizes relative to those observed in medium colony sizes. The Authors interpret this finding in two subtly different ways. The first is that Rho1 is a candidate regulator of egg size. This I agree with, although it is not surprising that Rho1 may be one of many factors regulating egg size given its role in cytoskeletal organization. The second interpretation is that "Rho1 as a key regulator of the active egg size adjustment of honey bee queens." This interpretation suggests that Rho1 is responsive to and "mediates" social cues from the queen and adjusts egg size accordingly. This would be a more exciting interpretation (one that would be worthy of publication in *eLife*), but in my view, the data do not support this second interpretation / conclusion. What I mean is that the data do not support a role for Rho1 as a 'plasticity gene' that mediates the responsiveness of eggs to social cues from the queen and then determines egg size according to colony size. If this were the case, I would have expected knockdown of Rho1 to reduce eggsize in small colonies and to INCREASE egg size in large colonies similar to that observed in medium colony sizes. Instead, Rho1 decreases egg size regardless of social circumstance, indicating that it only plays a role in regulating egg size, regardless of any social regulation. The Authors could have used their finding that egg size is reversible in response to colony size to see if knockdown of Rho1 would halt this reversibility. In summary, I do not believe that the Authors have sufficiently demonstrated that Rho1 "mediates" social regulation of egg size plasticity as their title suggests "Social Regulation of Egg Size Plasticity in the Honey Bee is Mediated by Cytoskeleton Organizer Rho1."

Major Concern # 2: I found the reported correlation between egg size and colony size very exciting and arresting, but this has already been published in this excellent article: Amiri, E., Le, K., Melendez, C.V., Strand, M.K., Tarpy, D.R., and Rueppell, O. (2020). Egg-size plasticity in *Apis mellifera*: Honey bee queens alter egg size in response to both genetic and environmental factors. Journal of Evolutionary Biology 33, 534-543. The results added to this manuscript (reversibility of egg size plasticity and induction of egg size plasticity in response to queen perception) are very interesting, but I would classify them as incremental additions.

Major Concern # 3: Finally, the evolutionary significance of egg size plasticity in response to colony size remains unclear. The Authors claim that these size differences are a result of differences in maternal provisioning. But can they really make such an interpretation without knowing if smaller eggs (large colonies) develop into smaller adult workers and if larger eggs (small colonies) develop into larger adult workers? In other social insects, like fire ants, the size of adult workers tends to increase with colony size. The logic is that as colonies increase in size, they bring more resources to the colonies and they can invest more in each individual. However, without knowing the final adult body sizes, it would be difficult to make strong inferences about what is happening exactly.

---

## [Author Response]

Essential revisions:1) Revise and qualify the conclusions on the role of Rho1 as a key regulator for egg size, in line with comments from Reviewers 2 and 3. Specifically, there is a need to better justify the idea that Rho1 is a "social mediator" of egg size as proposed in the title.

We agree that labelling Rho1 as a “social mediator” or “social regulator” would be inappropriate but this is not the same as stating that the social effect (of colony size) is acting through Rho1. We remain convinced that the multiple lines of evidence support the hypothesis that Rho1 is an important regulator of honey bee egg size. However, in response to the reviewers’ concerns, we have cautioned our language and conclusions throughout the manuscript, including the title.

2) Revise the section describing an effort to investigate the effects of egg-laying rate on egg size. To test this, you would need to compare queens in which egg-laying rate was experimentally increased or decreased to those where it was not. The presented manipulation instead compares queens in which egg-laying was ceased completely. If more appropriate data for this key question can be added please do so. Or the conclusions must be revised to better reflect the actual manipulation.

We have revised this section to clarify the experiment and limit our conclusions from it because we agree that the temporary halt of oviposition in our experiment might have a different effect than quantitative variation in oviposition rate. In addition, we have extracted additional data on the queens’ egg laying rate in our first experiment. These results show indeed a negative correlation between egg size and egg number in some circumstances but not in others. Thus, they complicate the interpretation of our findings and partially contradict our earlier study. However, we think that they are important to present in light of the reviewers’ concerns and for a comprehensive view of the reproductive behavior of honey bee queens. We have revised the discussion and conclusions accordingly.

3) A table or other system to allow readers to better keep track of the experimental manipulations of each queen and their use in the proteomic analyses (see Reviewer 2 comment 2).

We have revised Figure 1 to better explain the experimental manipulations and give the readers an intuitive and immediate overview of the experimental manipulations. We have also added clarification of which exact samples were used in the proteomic analysis. However, a comprehensive table or other system to label treatment and control groups across all experiments would not help readers in our opinion, as explained in more detail below (reviewer 2 comment 2). Instead, we like to keep the Materials and Methods and the Results separated in sections that correspond to the different experiments.

4) Greater clarity on the relationship between body size and egg size. You may follow the recommendation of Reviewer 2 (Comment 3): "Please add a scatter plot of wing size vs. baseline ovary size and wing size vs. baseline egg size (one point per queen). This will help your readership to better understand what is giving rise to the individual-level and species-level variation in egg and ovary size. Also, please add a few sentences to the Results to explain why you selected these traits to measure from each queen: that particular landmark to landmark distance on the wings, ovary weight, and this method of egg size measurement; also please include a sentence or two to summarize any important limitations of these selected measures."

Following this recommendation, we have added further justification for the chosen measures and discuss their limitations, following this suggestion. We have also investigated the relationship between body size (measured as wing size) and egg size. We conducted correlation analyses separately for eggs produced in small, medium, and large colonies and report these results now at the end of the first section in the “Results”. However, in light of our results that there is no relation between queen size and egg size (all correlations are non-significant), we do not think that an additional figure is warranted. The additionally suggested graph of baseline ovary size vs body size could not be generated because baseline ovary size could not be determined: The required destructive assay (dissection) was only performed at the end of the experiments, when queens were either in small or large colonies.

5) More data or justification on the fate of larger vs smaller eggs in honey bees, so that the likely ecological and evolutionary significance of the reported changes in egg size is clear.

Our initial discovery of the egg size plasticity in honey bees (Amiri et al. 2020) and work by other authors that we cite (e.g., Al-Kahtani and Bienefeld 2021) already document positive effects of large eggs. We have revised the Introduction and Discussion to alert the readers to these previous findings. The current study is focused on the causation of the egg size plasticity instead of the consequences, and we consider additional investigations of the consequences outside the scope of the study. For more detail, please see our reply to the 3^rd^ major concern of the 3^rd^ reviewer.

Reviewer #1 (Recommendations for the authors):This study makes several new contributions to our understanding of maternal control of egg size in honey bees. It seems reasonable to assume that larger eggs compensate in some way for smaller colonies (e.g. offset in some way the better provisioning given to newly-hatched larva in larger colonies?) but the actual significance of egg size is not much discussed in this study. I was left wondering whether larger worker eggs have been previously established to have higher survival in honey bees (if so, citations of those studies are needed) or whether this remains an assumed outcome of differences in egg size. Either way, I recommend more in the Intro or Discussion reflecting on the possible importance of egg size for fitness in the context of the honey bee life cycle.

We have added material to the introduction to inform the reader what is known about the consequences. We agree that there is potentially an adaptive explanation rooted in the colony life cycle, but think it would be premature to speculate about this hypothesis (as explained in our reply to the 3^rd^ major concern by reviewer #3 below).

Other comments:80: "..an adaptation to the colonial life cycle of honey bees but maybe more generally applicable to other oviparous animals." How might social cues affect oviparous animals that do not live in social groups?

Following the insinuated suggestion, we have revised this sentence.

86: Cite your previous study here when you refer to it.

We added the citation.

88: Give N here and elsewhere in Results when treatments are first introduced (several times I had to scroll down to Methods in order to find information about sample sizes).

The sample sizes can be directly inferred from the degrees of freedom in the ANOVA results reported. For other results, where df are not given, we have revised our manuscript to report sample sizes in the main text and figure captions.

119-126: I appreciate the effort here to experimentally investigate whether egg size is affected by egg-laying rate, but the question does not seem to be entirely answered. In this experiment, queens from large colonies are restricted from any egg-laying and then have their eggs measured again once egg-laying resumes. This is not really the same as simulating ongoing lower egg-laying rate (as might be experienced by a queen in a small colony). Your conclusion from this section is that egg size is not a by-product of higher or lower egg number, but is that saying that the two (number and size) are unrelated? Overall, I think the question of whether this important trade-off exists in honey bees warrants more discussion.

We agree with the reviewer that a quantitative restriction of egg laying would provide valuable additional insights and have added this statement to the discussion. We further agree that the two statements “egg number and size are uncorrelated” and “egg size is not a by-product of egg number” are not equivalent and have revised the discussion to clarify. We have also added additional data, which unfortunately do not resolve the issue unambiguously because a negative relation between number and size is observed in some instances but not all.

187: "All eggs appeared to be viable and differed phenotypically only in size" – was the viability of at least some eggs confirmed?

We did not follow the developmental fates of the eggs in this experiment because our size measurements involve the (destructive) removal from the brood cells. We have added this statement to the sentence to clarify.

Reviewer #2 (Recommendations for the authors):(1) Explanations for general readers. Given the broad readership of this journal, and the wide interest that there will be for this topic and your findings, we ask that you make the Results more approachable to readers who are not familiar with culturing honeybees. For instance at the beginning of the Results section, please give a general explanation of how these experiments are done. Some questions that a non-specialist reader will have: Can queens be moved between colonies that they did not found? When a queen was moved into a new colony, did the previous queen need to be removed first? What are sister queens, and is it important that sister queens be used for this sort of experiment? Also please give a few sentences of explanation to contextualize the categories of small, medium, and large colonies. How do these sizes compare to average/healthy colony sizes? Does the number of queens in a colony increase with colony size, is it controlled here?

Thank you. We have added the requested information in various places of the manuscript where we think it is most appropriately placed.

(2) Clarity in how you refer to treatments. I had some difficulty keeping track of the different treatments. Please add a table in which each row represents a distinct treatment. Make a unique text label for each unique treatment, and then give the relevant treatment details (E.g. "R1" : "Queen spent A days in colony of B size, then spent C days in colony of D size, then spent E days in colony of F size"). Then in the text you can use the labels to refer readers to the exact treatments ("We did proteomic analysis on queens from groups R1 and R2"), and then you can use more colloquial language to describe the results ("The ovaries that made large eggs were enriched with…"). Once this has been done, use these labels to make the figures clearer. Namely, in a given figure, all points that were from the same treatment group should be given the same color, and each individual queen from that treatment group should have a distinct symbol shape.

According to the recommendation, we have revised figure 1 to label the two treatments in this experiment clearly with two different colors. This experiment with its two treatments is the only complex experimental design. We have increased the clarity of treatments by adding more “methods” to the Results section and improving the correspondence between the Materials and methods and the Results sections. However, the suggested table to summarize all methods is impossible to implement. For example, the proteomic comparison was performed between queens housed in small and large colonies (producing large and small eggs correspondingly), mixing queens from the first experiment (that had been transferred repeatedly) and queens from the second experiment (with only one transfer) to increase sample size. How long these queens were kept in which colony is described in the Methods section and is also depicted on the x-axis of Figure 1, 3, 4, and 8, but a letter designation (as suggested) is not possible.

(3) Wing size, ovary size, and interindividual variation. The data that have been collected-wing size, ovary size, and egg size for all the queens-can help to shed light on sources of egg size variation. Wing size cannot change during adulthood and so it can be treated as a measure of overall size/nutrition during initial queen maturation, while ovary size and egg size are the factors that respond to the environmental condition. Please add a scatter plot of wing size vs. baseline ovary size and wing size vs. baseline egg size (one point per queen). This will help your readership to better understand what is giving rise to the individual-level and species-level variation in egg and ovary size. Also, please add a few sentences to the Results to explain why you selected these traits to measure from each queen: that particular landmark to landmark distance on the wings, ovary weight, and this method of egg size measurement; also please include a sentence or two to summarize any important limitations of these selected measures.

We have added additional analyses to show that individual body size variation is unrelated to egg size. In light of the non-significant results, we do not think that an additional figure is warranted. We have added justification for each of the measures in the Methods section.

(4) How does the shape of eggs change when egg size changes?

We did not notice any size-dependent shape changes. However, we did not specifically investigate this and therefore cannot speculate on possibly subtle allometries.

(5) Do you have data on the number of ovarioles for any of the queens? If so, please add a plot showing the relationship between ovariole number and baseline egg size.

We did not determine the ovariole numbers of the queens involved in this study. To the best of our knowledge, the ovariole number does not change during adult life and thus this trait was not considered important for egg size plasticity, the key aspect of our study. We understand that this trait is very important in comparative context (e.g., Church et al. 2021), social insect caste polymorphism (Wilson 1971) and social organization of honey bee workers (Page et al. 2012).

(6) This is a more detailed explanation for a comment I made in the Public Review. The experiment that is described as studying "egg laying rate" does not really address egg laying rate. As I understand it, the queens are moved to a place where there are no suitable places to lay eggs. It seems that the authors are saying that even if the queen can't lay eggs, the eggs don't just get bigger and bigger in the ovaries while they wait, and this is evidence that there isn't a relationship between laying rate and size. Based on the existing literature, I don't think there is any reason to think that restricting oviposition would or could cause an increase in egg size. In general, when insects don't have a good oviposition site, they pause oogenesis, resorb the eggs, or lay the eggs anyway (in a non-suitable oviposition site). The cleaner way to ask if there is a relationship between egg size and laying rate is to start by quantifying it in your size-manipulation experiments. Why not just count how many eggs the queens are laying per [unit of time] in the big and small colonies? If you do not have the data for that, that's fine, but that is certainly what researchers generally do when studying egg laying rate. Bottom line: this experiment does answer some other interesting questions, but not about egg-laying rate. Please re-frame this section in terms of "restricted oviposition" or some other phrase that you prefer.

We thank the reviewer for this suggestion and have followed it by retrieving the data of how many eggs were actually produced in the first experiment while queens were housed in different sized colonies. In contrast to our previous study, these data show indeed a negative relation between egg size and egg number, at least in 3 out of the 6 experimental weeks. We have added this result and modified the manuscript accordingly. However, we want to note here that the negative relation between the two does not mean that they are causally linked. Instead, they could be co-regulated by an upstream process. In this context, the experimental restriction of oviposition is a valuable addition because this experiment shows that egg laying rate and egg size can be decoupled even if this occurs under unnatural circumstances.

(7) Please be sure sample sizes are included throughout the text and in figure legends (egg measurements and individual queens), and in the new table recommended in Major Point (2).

In the text, we now specify sample sizes when they cannot be inferred from the degrees of freedom in the test statistics. We also now indicate sample sizes in the legend for figures 2 and 8, where summary statistics instead of individual data points are shown.

(8) Include some discussion of the stages of oogenesis in the Results and/or Discussion. When an adult is increasing or decreasing the size of the eggs she lays, this is most likely happening by changing the amount of time that the developing eggs spend in one or more stages of oogenesis (and the leading candidate would be the period of vitellogenesis, presumably). This is also the simplest explanation for the differences in protein expression levels. In the sections on proteomics, gene expression patterns, and/or RNAi, consider if any discussion of this topic might be informative and if so include it. It is probably beyond the scope of this project to carefully analyze the process of oogenesis that gives rise to the various sizes, but that would be a very interesting future direction. On a related point, I am quite confused by the ovary weight finding. How does the ovary weigh less with bigger eggs? Are there fewer eggs? That is what I assumed but I could be wrong. Please include a bit more discussion of this surprising finding, in the Results or Discussion.

We agree that vitellogenesis is the most likely period in oogenesis to affect egg size and have now added this statement to the discussion. Whether the effect is achieved by a temporal delay during maturation, physiological upregulation of metabolism and transport mechanisms, or other regulatory processes is speculative, but we think that our proteomic results suggest a physiological upregulation instead of a retardation of the egg during vitellogenesis. Additional follow-up studies are needed…

(9) I am excited about the authors' idea that this is life history without the nutrition limitations, but it will be helpful to give readers a little information about queen nutrition (in the Results and/or Discussion). How would her amount of food differ between small and large colonies? Do 10x more workers means she eats 10x as much?

We agree that this is an exciting research topic. Unfortunately, not much is known yet and we have now included that statement in the text to stimulate further research.

(10) A few relevant sources to cite. There is another instance of plastic changes to eggs during insect oogenesis in response to perception: Abram, Paul K., et al. "An insect with selective control of egg coloration." Current Biology 25.15 (2015): 2007-2011. Also there was a recent effort to determine if egg size is related to egg number (or other things), and it does include social insects (see lines 41-42): Church, Samuel H., et al. "Repeated loss of variation in insect ovary morphology highlights the role of development in life-history evolution." Proceedings of the Royal Society B 288.1950 (2021): 20210150.

We thank the reviewer for those suggestions. The plastic modulation of egg coloration by stinkbugs is certainly an exciting finding but we had trouble finding a relevant connection to our study and thus did not incorporate it. However, the Church et al. study is highly relevant (but wasn’t published when our manuscript was first written) and thus is now cited to put our study into this broader context.

Reviewer #3 (Recommendations for the authors):Two major conclusions are drawn in the manuscript by Han et al.: (1) that queens predictably and reversibly increase egg size in small colonies and decrease egg size in large colonies, while their ovary size changes in the opposite direction; and (2) that Rho1 is a candidate regulator of egg size. The Authors interpret these two findings in the following way (Lines 201 and 202): "we identify the cytoskeleton organizer Rho1 as a key regulator of the active egg size adjustment of honey bee queens." While I admire the work that Authors have done here, I have the following major concerns:Major Concern # 1: The authors knock down Rho1 using RNAi and find that it reduces egg size in both large and small colony sizes relative to those observed in medium colony sizes. The Authors interpret this finding in two subtly different ways. The first is that Rho1 is a candidate regulator of egg size. This I agree with, although it is not surprising that Rho1 may be one of many factors regulating egg size given its role in cytoskeletal organization. The second interpretation is that "Rho1 as a key regulator of the active egg size adjustment of honey bee queens." This interpretation suggests that Rho1 is responsive to and "mediates" social cues from the queen and adjusts egg size accordingly. This would be a more exciting interpretation (one that would be worthy of publication in eLife), but in my view, the data do not support this second interpretation / conclusion. What I mean is that the data do not support a role for Rho1 as a 'plasticity gene' that mediates the responsiveness of eggs to social cues from the queen and then determines egg size according to colony size. If this were the case, I would have expected knockdown of Rho1 to reduce eggsize in small colonies and to INCREASE egg size in large colonies similar to that observed in medium colony sizes. Instead, Rho1 decreases egg size regardless of social circumstance, indicating that it only plays a role in regulating egg size, regardless of any social regulation. The Authors could have used their finding that egg size is reversible in response to colony size to see if knockdown of Rho1 would halt this reversibility. In summary, I do not believe that the Authors have sufficiently demonstrated that Rho1 "mediates" social regulation of egg size plasticity as their title suggests "Social Regulation of Egg Size Plasticity in the Honey Bee is Mediated by Cytoskeleton Organizer Rho1."

We thank the reviewer for the clarification, understand the distinction made by the reviewer, and agree that we need to be more precise about which conclusions can and which cannot be drawn based on our results. However, we do not understand the argument that Rho1 knockdown should increase egg size in large colonies and reduce egg size in small colonies to be involved in egg size determination. In our view such context-dependency could be a possibility but is not a requirement of Rho1 regulating egg size in response to colony size. Nevertheless, we have generally cautioned our conclusions with regard to Rho1 as explained above.

Major Concern # 2: I found the reported correlation between egg size and colony size very exciting and arresting, but this has already been published in this excellent article: Amiri, E., Le, K., Melendez, C.V., Strand, M.K., Tarpy, D.R., and Rueppell, O. (2020). Egg-size plasticity in Apis mellifera: Honey bee queens alter egg size in response to both genetic and environmental factors. Journal of Evolutionary Biology 33, 534-543. The results added to this manuscript (reversibility of egg size plasticity and induction of egg size plasticity in response to queen perception) are very interesting, but I would classify them as incremental additions.

We thank the reviewer for the appreciation of our initial discovery. However, we disagree with the assessment that the reported results are just an incremental addition to our previous study because we report several important and novel findings.

Major Concern # 3: Finally, the evolutionary significance of egg size plasticity in response to colony size remains unclear. The Authors claim that these size differences are a result of differences in maternal provisioning. But can they really make such an interpretation without knowing if smaller eggs (large colonies) develop into smaller adult workers and if larger eggs (small colonies) develop into larger adult workers? In other social insects, like fire ants, the size of adult workers tends to increase with colony size. The logic is that as colonies increase in size, they bring more resources to the colonies and they can invest more in each individual. However, without knowing the final adult body sizes, it would be difficult to make strong inferences about what is happening exactly.

We agree with the reviewer that the evolutionary significance of the plasticity (or its adaptive value) is important. In our previous work that is cited by the reviewer above, we found that a larger egg size is associated with reduced offspring mortality. Work by other authors that we cite in the manuscript relates egg size to caste, with larger eggs more likely developing into queens and also into “better” queens. Both of these lines of evidence indicate a beneficial effect of egg size but more studies on the consequences are certainly needed. The study that we present here is focused on the causation and not additional consequences of egg size variation. Thus, we rely on our previous results and inferences from the literature that large eggs are generally advantageous. Our operational hypothesis of the evolutionary significance is that the reported egg size plasticity is an adaptation to the annual colony cycle of honey bees that entails colony size fluctuations and reproductive swarming. However, we find it too speculative to elaborate on these ideas here in this manuscript, which is focused on the proximate causes of the plasticity instead. In contrast to fire ants and some other social insects, the workers of honey bees are quite uniform in size and therefore adult size variation within the worker caste seems an unlikely evolutionary explanation. However, we cannot rule it out and shall investigate this hypothesis in future studies on the consequences of egg size variation in honey bees.